# Akt regulation of glycolysis mediates bioenergetic stability in epithelial cells

Yin P Hung[1,2,3†], Carolyn Teragawa[4†], Nont Kosaisawe[4], Taryn E Gillies[4], Michael Pargett[4], Marta Minguet[4], Kevin Distor[4], Briana L Rocha-Gregg[4], Jonathan L Coloff[1], Mark A Keibler[5], Gregory Stephanopoulos[5], Gary Yellen[2], Joan S Brugge[1], John G Albeck[4]*

[1]Department of Cell Biology, Harvard Medical School, Boston, United States; [2]Department of Neurobiology, Harvard Medical School, Boston, United States; [3]Department of Pathology, Brigham and Women's Hospital, Boston, United States; [4]Department of Molecular and Cellular Biology, University of California, Davis, United States; [5]Department of Chemical Engineering, Massachusetts Institute of Technology, Cambridge, United States

**Abstract** Cells use multiple feedback controls to regulate metabolism in response to nutrient and signaling inputs. However, feedback creates the potential for unstable network responses. We examined how concentrations of key metabolites and signaling pathways interact to maintain homeostasis in proliferating human cells, using fluorescent reporters for AMPK activity, Akt activity, and cytosolic NADH/NAD$^+$ redox. Across various conditions, including glycolytic or mitochondrial inhibition or cell proliferation, we observed distinct patterns of AMPK activity, including both stable adaptation and highly dynamic behaviors such as periodic oscillations and irregular fluctuations that indicate a failure to reach a steady state. Fluctuations in AMPK activity, Akt activity, and cytosolic NADH/NAD$^+$ redox state were temporally linked in individual cells adapting to metabolic perturbations. By monitoring single-cell dynamics in each of these contexts, we identified PI3K/Akt regulation of glycolysis as a multifaceted modulator of single-cell metabolic dynamics that is required to maintain metabolic stability in proliferating cells.
DOI: https://doi.org/10.7554/eLife.27293.001

*For correspondence: jgalbeck@ucdavis.edu

†These authors contributed equally to this work

Competing interests: The authors declare that no competing interests exist.

## Introduction

A central function of cellular metabolic regulation is to ensure an adequate supply of metabolites for bioenergetics and biosynthetic processes. To maintain metabolic homeostasis, cells utilize feedback loops at multiple levels in an integrated metabolic-signaling network. For instance, glycolysis is regulated by feedback control at the level of phosphofructokinase, which senses the availability of ATP and the respiratory intermediate citrate. Additionally, in response to ATP depletion, the energy-sensing kinase AMP-activated protein kinase (AMPK) stimulates glucose uptake and suppresses energy-consuming processes (*Hardie, 2008*). These homeostatic pathways respond to bioenergetic stress by increasing or decreasing the appropriate metabolic fluxes to return the cell to a state with stable and sufficient levels of key metabolites. While bioenergetic stress can occur when any of a number of metabolites becomes critically limited, we focus in this study on the key metabolite ATP because of its broad importance as an energy source for cellular processes, and because AMPK activity can be used as a reliable indicator of low ATP:AMP ratios within the cell. We therefore use the term bioenergetic stress here to indicate a situation in which the concentration of available ATP is reduced, as indicated by AMPK activation.

Bioenergetic stress can result from a loss of ATP production, such as when nutrients become limited or metabolic pathways are inhibited by a pharmacological agent. Alternatively, ATP depletion can also result from an increase in ATP usage, such as when anabolic processes are engaged during cell growth. Because anabolic processes such as protein translation can use a large fraction (20–30%) of cellular ATP (*Buttgereit and Brand, 1995*; *Rolfe and Brown, 1997*), it is unsurprising that cellular proliferation and metabolic regulation are tightly linked (*Gatenby and Gillies, 2004*; *Wang et al., 1976*). Growth factor (GF) stimulation activates the PI3K/Akt pathway, which plays a key role in proliferation by stimulating both cell cycle progression and mTOR activity, leading to increased protein translation. Simultaneously, Akt activity promotes glucose metabolism by stimulating the activity of hexokinase (*Roberts et al., 2013*) and phosphofructokinase (*Novellasdemunt et al., 2013*) and translocation of glucose transporters (Glut1 and Glut4) to the cell surface (*Sano et al., 2003*; *Wieman et al., 2007*), while PI3K enhances the activity of hexokinase, phosphofructokinase, and aldolase to increase glycolytic flux (*Hu et al., 2016*; *Inoki et al., 2012*; *Inoki et al., 2003*).

The balance of anabolic and catabolic processes is particularly important in epithelial tissues, as they maintain the capacity to proliferate throughout adult life. Most cancers arise in epithelial cells (*Koppenol et al., 2011*) and involve a loss of both signaling and metabolic regulation (*Gwinn et al., 2008*; *Vander Heiden et al., 2009*). The AMPK and Akt pathways play key roles in this balance, intersecting through multiple crosstalk points and feedback loops to control both glucose metabolism (*Figure 1—figure supplement 1*) and protein translation at the level of mTOR. In principle, an optimal feedback response to an ATP-depleting perturbation would allow ATP to rapidly increase and stabilize at a sufficient level, while unstable responses such as continuing fluctuations or oscillations could be deleterious for the cell. However, a system with multiple feedbacks requires unavoidable tradeoffs in efficiency and robustness, and feedback increases the potential for instability (*Chandra et al., 2011*). Experimentally, such unstable metabolic responses have been observed in yeast (*Danø et al., 1999*; *Ghosh and Chance, 1964*) and in specialized post-mitotic mammalian cell types (*Chou et al., 1992*; *O'Rourke et al., 1994*; *Tornheim and Lowenstein, 1973*; *Yang et al., 2008*), confirming the potential for instability during metabolic adaptation. However, in epithelial cells, little is known about the kinetic relationships between signaling and metabolic activity that allow proliferation and other anabolic processes to proceed in step with energy production.

To understand the kinetics of homeostasis, single-cell data are needed because of the potential for metabolic state to vary even among genetically identical cells. Events that are asynchronous among cells, and subpopulations with differential behaviors, are not apparent in the population mean due to their tendency to 'average out' (*Purvis and Lahav, 2013*). Until recently, dynamics in metabolism could only be measured effectively under conditions where fluctuations are synchronized across populations of cells, because biochemical techniques such as mass spectrometry provide broad measurement capabilities but reflect the population average rather than individual cells. However, advances in fluorescent reporters now enable real-time monitoring of metabolic and signaling dynamics in individual intact cells. Genetically encoded fluorescent protein-based reporters have been designed to respond to specific metabolites by changes in their fluorescence output (*Tantama et al., 2012*; *Tsien, 2005*). As a result, metabolic and signal transduction states, including cytosolic NADH-NAD$^+$ ratio (*Hung et al., 2011*; *Zhao et al., 2015*), glutathione redox potential (*Gutscher et al., 2008*), ATP-ADP ratio (*Berg et al., 2009*; *Tantama et al., 2013*), and AMPK activity (*Tsou et al., 2011*), can now be monitored in living cells.

In this study, we established a panel of proliferative epithelial cells expressing multiple fluorescent biosensors to enable detailed tracking of single-cell metabolic responses. We first used pharmacologic compounds to induce bioenergetic stress and to establish the range of cellular responses, finding that different forms of metabolic inhibition trigger strikingly different kinetics of adaptation and reveal conditions under which stable adaptation fails. We then used this framework to examine how metabolic adaptation functions in proliferating cells. We found that periods of bioenergetic stress occur throughout the normal cell cycle, and we identified a prominent role for PI3K/Akt regulation of glycolysis in mediating metabolic stability at the single-cell level.

## Results

### Fluorescent reporters enable single-cell imaging of metabolic and signaling dynamics

MCF10A mammary epithelial cells are dependent on GF stimulation for proliferation and can be stimulated to proliferate at different rates (*Ram et al., 1995*), making them a useful experimental system to examine the relationship between proliferation and metabolic kinetics. We generated MCF10A cell lines stably expressing a panel of genetically encoded fluorescent reporters for central components of the metabolic-signaling control network. First, the sensor AMPKAR2 (*Figure 1A*) was constructed to monitor the activity of AMPK. Using the FRET-based AMPKAR biosensor (*Tsou et al., 2011*), we improved its dynamic range by using a brighter donor fluorescent protein mTurquoise2 (*Goedhart et al., 2012*) and an extended 'EV' linker (*Komatsu et al., 2011*). In live cells, AMPK activation was calculated based on a linear ratiometric method (*Birtwistle et al., 2011*), which we term 'AMPK index' (see Materials and methods for details of activity calculations). We verified the sensing capability of AMPKAR2 by treating MCF10A-AMPKAR2 cells with the direct AMPK activator AICAR (*Figure 1B and C*). Following AICAR application, all cells showed an immediate increase in the AMPK index and reached a steady state within 5 hr. When cultured in the absence of glucose, pyruvate, and glutamine for 24 hr, MCF10A-AMPKAR2 cells showed elevated AMPK index, which decreased abruptly upon glucose addition (*Figure 1D*). Cells cultured in combinations of glucose, pyruvate, or glutamine displayed varying elevated levels of steady-state AMPK index (*Figure 1—figure supplement 2A–B*), demonstrating that AMPKAR2 could monitor AMPK status across a range of physiological concentrations of nutrients in individual live cells.

To assess the dynamics of the cytosolic NADH-NAD$^+$ redox state, we utilized the fluorescent biosensor Peredox, which is based on a circularly permuted green fluorescent protein T-Sapphire conjugated to the bacterial NADH-binding protein Rex (*Hung et al., 2011*). To maintain compatibility with red-wavelength reporters for dual imaging and to simplify cell tracking, we generated a nuclear-targeted Peredox fused to the YFP mCitrine (*Figure 1E*). NADH is a major redox cofactor in glycolysis, which generates NADH from NAD$^+$ via the glyceraldehyde-3-phosphate dehydrogenase (GAPDH) reaction in the cytosol. As NADH and NAD$^+$ exchange freely between nuclear and cytosolic compartments, Peredox nuclear signal reports the cytosolic NADH-NAD$^+$ redox state and serves as an indicator of glycolytic activity (*Hung et al., 2011*). Once normalized by the fused mCitrine signal to correct for variations in biosensor expression, Peredox nuclear signal is thus defined as the 'NADH index.' To verify cytosolic NADH-NAD$^+$ redox sensing, we exploited the lactate dehydrogenase reaction,which interconverts pyruvate and lactate with concomitant exchange of NADH for NAD$^+$. MCF10A-Peredox cells treated with lactate, or pyruvate in combination with iodoacetate (IA) as a glycolytic blockade, exhibited rapid (<3 min) changes to reach maximal and minimal sensor responses, respectively (*Figure 1F and G*), indicative of cytosolic NADH-NAD$^+$ redox sensing. Consistent with previous data (*Hung et al., 2011*), steady-state NADH index increased with glucose concentrations (*Figure 1—figure supplement 2C*), and Peredox could detect glycolytic dynamics in individual live cells (*Figure 1H*). A control reporter with a mutation in the NADH-binding site (Y98D) predicted to abrogate NADH binding failed to respond to the same conditions (*Figure 1—figure supplement 3*).

To track PI3K/Akt pathway activity, we constructed a reporter based on the Forkhead transcription factor FOXO3a. Akt phosphorylation of FOXO3a promotes its cytoplasmic retention; with low Akt activity, dephosphorylated FOXO3a translocates to the nucleus (*Brunet et al., 1999*; *Tran et al., 2002*). To monitor Akt activity, we thus fused a red fluorescent protein mCherry to a truncated FOXO3a gene in which transcriptional activity was abrogated to minimize any interference on endogenous gene transcription, a strategy previously shown to specifically report Akt activity (*Gross and Rotwein, 2016*; *Maryu et al., 2016*). We refer to this construct here as AKT-KTR; the Akt activity indicated by its cytosolic-to-nuclear fluorescence ratio is referred to as the 'Akt index' (*Figure 1I*). Upon insulin application following GF deprivation, MCF10A-AKT-KTR cells showed an abrupt increase in Akt index (*Figure 1K*), consistent with the expected Akt stimulation. Conversely, pharmacologic treatment with the Akt inhibitor MK2206 induced an immediate decrease in Akt index (*Figure 1J and L*), confirming that AKT-KTR could report Akt activity dynamics in individual live cells. We generated dual-reporter cell lines expressing both AKT-KTR and AMPKAR2, or AKT-KTR and Peredox, allowing Akt activity and metabolic status to be measured in the same cell. To

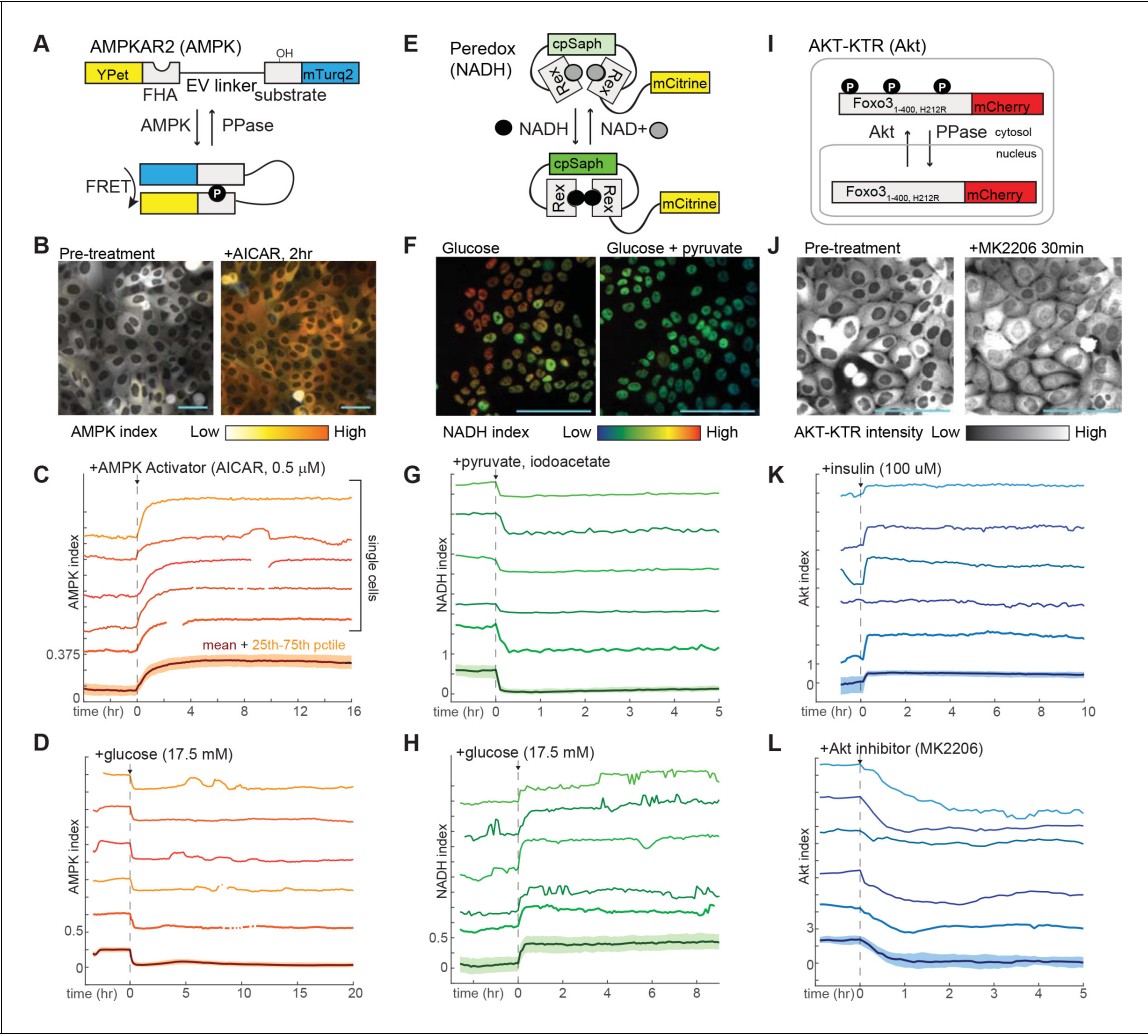

**Figure 1.** Design and validation of live-cell reporters for AMPK (**A–D**), cytosolic NADH/NAD$^+$ ratio (**E–H**), and Akt kinase activity (**I–L**). A,E, and I depict schematic diagrams for each of the reporters. B,F, and J show representative microscope images of MCF10A cells stably expressing each reporter. In images in B and F, reporter activity is represented as a pseudocolored image calculated as a ratio of the reporter components (see Materials and methods for details of calculations). C, D, G, H, K, and L display individual and aggregate cell data for the indicated treatments. For each panel, the bottom-most profile represents the mean measurement for a population of >200 cells; the colored region around the mean indicates the 25th to 75th percentile range for the population. The five traces above the mean plot depict five representative individual cells, plotted at the same scaling as the mean. The media used in each experiment (prior to the indicated additions) were as follows: C – iGM (imaging-modified growth medium; see Materials and methods); D – iGM lacking glucose; G – iGM2; H – iGM lacking glucose; K – iGM lacking insulin; L – iGM.

DOI: https://doi.org/10.7554/eLife.27293.002

The following figure supplements are available for figure 1:

**Figure supplement 1.** Schematic diagram of feedbacks connecting ATP production with glycolytic regulation.
DOI: https://doi.org/10.7554/eLife.27293.003

**Figure supplement 2.** Measurements of AMPK and NADH indices under different nutrient conditions.
DOI: https://doi.org/10.7554/eLife.27293.004

**Figure supplement 3.** Lack of response of a mutant control NADH reporter to stimuli known to control cellular NADH levels.
DOI: https://doi.org/10.7554/eLife.27293.005

**Figure supplement 4.** Comparison of metabolic properties of parental and reporter cell lines.
DOI: https://doi.org/10.7554/eLife.27293.006

track the relationship between proliferation and metabolic analysis, we also constructed a cell line expressing both AMPKAR2 and a reporter of S/G2 phase, GMNN-mCherry (*Albeck et al., 2013*; *Sakaue-Sawano et al., 2008*). In comparison to parental cells, reporter-expressing cells did not differ substantially in bulk metabolic properties, including basal oxygen consumption and lactate secretion, with the exception of AMPKAR2/AKT-KTR cells, which had a higher maximal glycolytic rate (*Figure 1—figure supplement 4*).

We used our reporter cell lines to establish the relationship between metabolic, signaling, and cell cycle parameters under different GF stimuli. Consistent with previous studies (*Worster et al., 2012*) insulin treatment induced the strongest activation of Akt index, while epidermal growth factor (EGF) produced a more moderate activation (*Figure 2A*). Glucose uptake and NADH index were also highest in insulin-treated cells, intermediate in EGF-treated cells, and lowest in the absence of

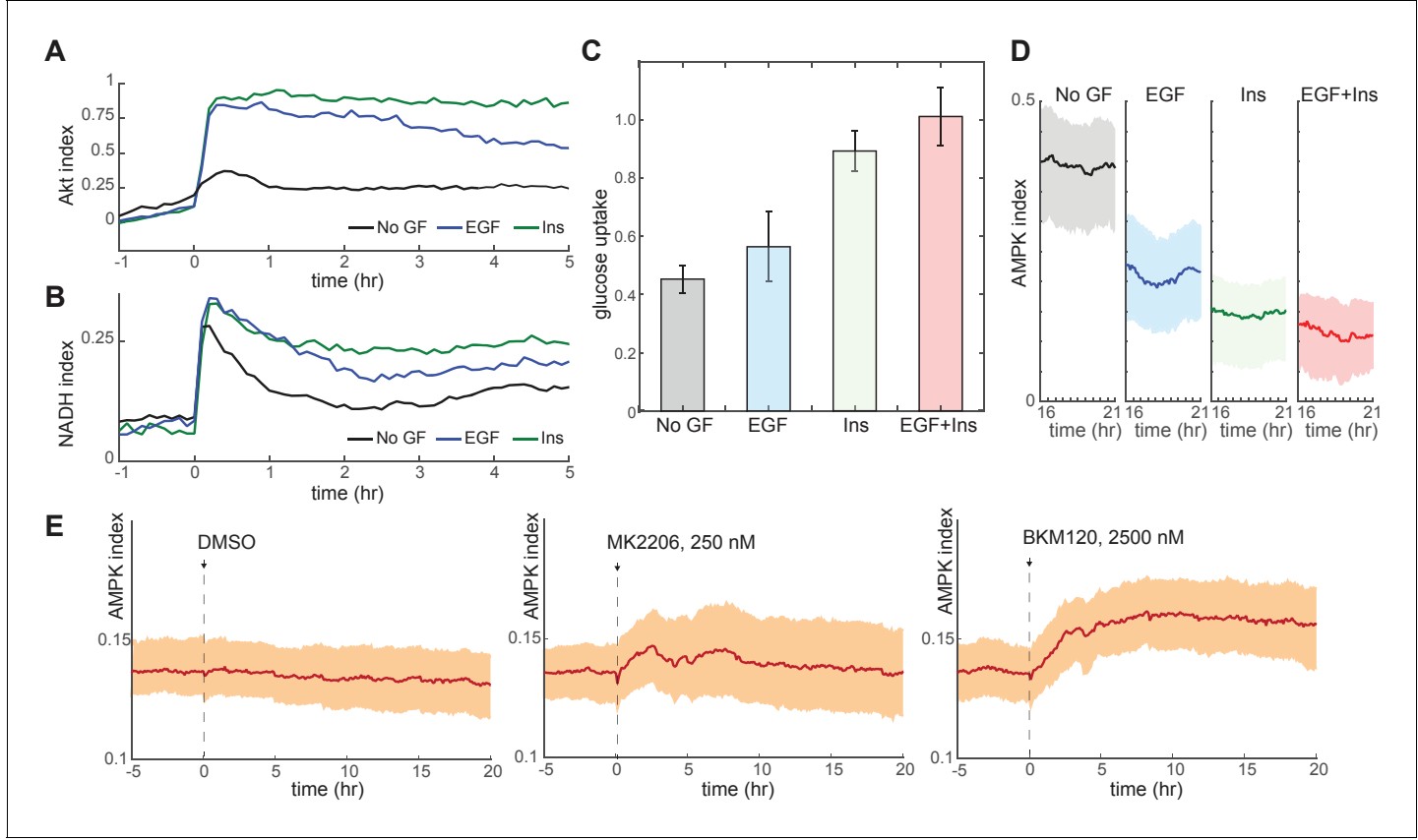

**Figure 2.** Glucose metabolism drives adaptation to the bioenergetic stress of proliferation. (**A**) Mean Akt index, measured by AKT-KTR, following stimulation with EGF or insulin. Prior to imaging, cells were placed in medium lacking EGF or insulin. GFs were added at time 0. $N = 4$, combined. (**B**) Mean NADH index, measured by Peredox, following stimulation with EGF or insulin. Prior to imaging, cells were placed in medium lacking EGF or insulin. GFs were added at time 0. $N = 4$, combined. (**C**) Glucose uptake from culture medium by MCF10A cells. Glucose depletion from the medium was assayed immediately following a 2-hr period during which the cells were exposed to the indicated conditions. Glucose uptake is shown relative to EGF +insulin treated cells, to which other conditions were normalized. Bars represent the average, and error bars the standard deviation, of four independent clones measured in triplicate in one experiment; results are representative of 3 total experiments run on different days. (**D**) Mean and 25th-75th percentile AMPK index for cells treated as in (**A**), recorded between 16 and 21 hr following treatment. $N = 5$, representative. (**E**) Mean AMPK index measurements for MCF10A cells cultured in medium containing glucose, glutamine, and insulin and treated at time 0 with either DMSO, 250 nM MK2206, or 2.5 μM BKM120. $N = 4$, representative.

DOI: https://doi.org/10.7554/eLife.27293.007

The following figure supplements are available for figure 2:

**Figure supplement 1.** Changes in proliferation and metabolism under conditions affecting PI3K/Akt signaling.
DOI: https://doi.org/10.7554/eLife.27293.008
**Figure supplement 2.** Single-cell variability in NADH and AMPK indices.
DOI: https://doi.org/10.7554/eLife.27293.009

GFs (*Figure 2B and C*), while the average AMPK index correlated inversely with Akt index (*Figure 2D*). Because EGF stimulated proliferation more strongly than insulin (*Figure 2—figure supplement 1A*), these metabolic parameters correlated poorly with proliferative rate. Together, these results suggest that the increased rates of glucose uptake and metabolism stimulated by Akt activity are the primary determinants of metabolic status under each GF. Accordingly, treatment of insulin-cultured cells with an Akt or PI3K inhibitor decreased glucose uptake and NADH index (*Figure 2—figure supplement 1B–C*) while increasing AMPK index (*Figure 2E*). When NADH and AMPK index were tracked in individual cells, we found a high degree of variance over time within each cell, with some cells showing pronounced peaks and troughs (*Figure 2—figure supplement 2*). While this behavior appeared to correlate with certain GF conditions, we lacked a framework to quantify and interpret these dynamics; we therefore turned to defined metabolic perturbations as a tool to first establish basic homeostatic responses for single cells.

## Fluctuating single-cell responses to metabolic challenges

To assess the range of individual cellular responses to specific bioenergetic challenges, we exposed MCF10A-AMPKAR2 cells to a panel of metabolic inhibitors, including oligomycin (an inhibitor of the mitochondrial F0/F1 ATPase), carbonyl cyanide m-chlorophenyl hydrazonesodium (CCCP, a mitochondrial proton gradient uncoupler), and IA(an alkylating agent that inhibits the glycolytic enzyme GAPDH with minimal effects on other cellular thiols at <100 µM [*Schmidt and Dringen, 2009*]). As expected, each of these compounds rapidly raised the mean AMPK index in a dose-dependent manner in cells cultured in growth medium, confirming that they perturb ATP homeostasis in proliferating MCF10A cells (*Figure 3* and *Videos 1–3*). However, each inhibitor induced strikingly different kinetics at the single-cell level. Notably, IA- induced periodic oscillations of AMPK index, most evident at intermediate (5–10 µM) IA concentrations in which oscillations were sustained for as many as 50 cycles over 20 hr (*Figure 3A*). These fluctuations of AMPK activity were not synchronized among individual cells and thus not apparent in the population average measurements. In contrast, oligomycin induced an immediate increase in AMPK index that peaked at ~40 min but then fell, followed by a series of irregular pulses of AMPK activity ranging in duration from 1 to 3 hr (*Figure 3B*).

The highly dynamic nature of AMPK responses at the single-cell level was unexpected, and we therefore investigated whether these effects were a consequence of the particular imaging methods or experimental systems used. The asynchronous nature of these fluctuations argued against imaging artifacts or environmental fluctuations, which would affect all cells simultaneously. Similar kinetics were observed at all reporter expression levels, with moderate trends toward higher pulse frequency and shorter pulse duration in cells with higher reporter expression (*Figure 3—figure supplement 1*). Immunofluorescence detection of phosphorylated ACC (pACC), an AMPK substrate, showed indistinguishable staining patterns in parental and reporter-expressing cells (*Figure 3—figure supplement 2*). Importantly, the large cell-to-cell variation in pACC staining pattern observed by immunofluorescence in cells fixed at later time points is exactly what would be expected given the asynchronous fluctuations observed in live-cell experiments. Similar kinetic patterns were also observed in other mammary epithelial cell lines (184A1 and MCF12A; *Figure 3—figure supplement 3A*), indicating that they are not specific to the MCF10A cell line. Finally, the kinetics of feedback responses were common to the point of inhibition, as other inhibitors of the electron transport chain including metformin and antimycin A induced pulses with durations similar to those of oligomycin (*Figure 3—figure supplement 3B*). Together these data indicate that the kinetics observed are not narrowly confined to specific experimental conditions and may provide general insight into metabolic regulation in epithelial cells.

Because pulsatile AMPK activity was a common feature of the single-cell response to multiple perturbations, we developed a 'pulse score' to quantify the cumulative intensity of fluctuations for each cell over time (*Figure 3—figure supplement 4* and Materials and methods section). Oligomycin and IA-treated cells showed significantly increased pulse scores relative to untreated cells (*Figure 3D*). However, other perturbations resulted in different kinetic variations; for instance, at low and intermediate doses, CCCP induced a rapid increase in AMPK index, with a magnitude comparable to the other perturbations, that was maintained at a steady-state level for many hours with a low pulse score (*Figure 3C and D*). Higher doses of CCCP, which are known to inhibit respiration, exhibited similar effects as oligomycin. We speculate that, at concentrations where CCCP acts only as an ionophore, a new stable steady state is reached due to ATP consumption by the F0/F1 ATPase

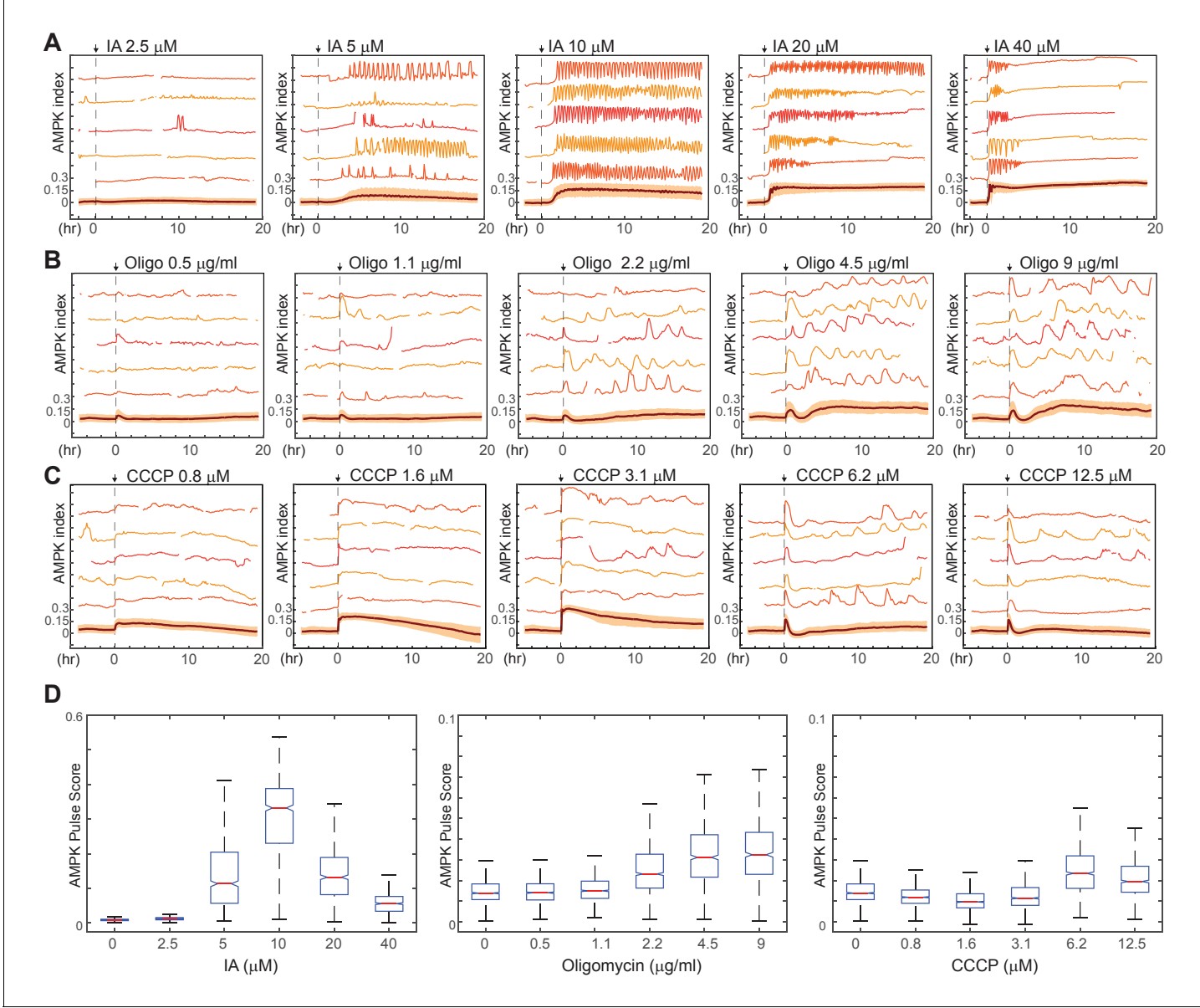

**Figure 3.** Kinetics of AMPK response to chemically induced metabolic stresses. (**A–C**) Prior to imaging, MCF10A-AMPKAR2 cells were placed in iGM, and were then treated during imaging at time 0 with various concentrations of IA (**A**), oligomycin A (**B**), or CCCP (**C**). Mean and representative single-cell traces are shown as in *Figure 1*. N = 3, representative. (**D**) Fluctuation scores for each of the conditions shown in (**A-C**), calculated as described in the Materials and methods section. Scores were calculated for the period beginning 1 hr after treatment (to exclude the initial peak) and continuing through the end of the experiment.

DOI: https://doi.org/10.7554/eLife.27293.010

The following figure supplements are available for figure 3:

**Figure supplement 1.** Comparison of drug-induced kinetics as a function of reporter expression.

DOI: https://doi.org/10.7554/eLife.27293.011

**Figure supplement 2.** Comparison of drug-induced AMPK signaling in parental and reporter-expressing cells by immunofluorescence.

DOI: https://doi.org/10.7554/eLife.27293.012

**Figure supplement 3.** AMPK index responses to other electron transport chain inhibitors and in other cell lines.

DOI: https://doi.org/10.7554/eLife.27293.013

**Figure supplement 4.** Identification of peaks and quantification of fluctuation trends in time-dependent reporter signals.

DOI: https://doi.org/10.7554/eLife.27293.014

**Figure supplement 5.** Cell cycle responses during treatment with metabolic challenges.

DOI: https://doi.org/10.7554/eLife.27293.015

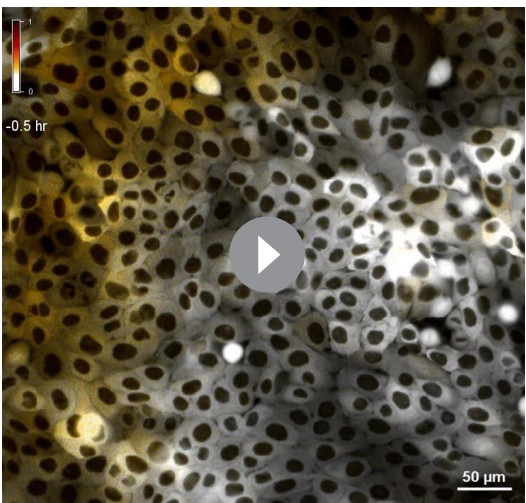

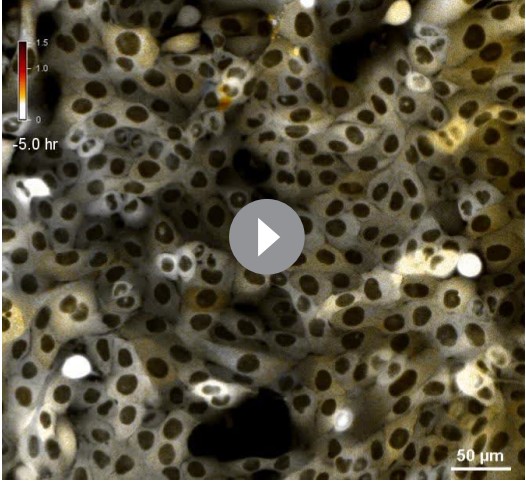

**Video 1.** Induction of AMPK oscillations by inhibition of glycolysis. MCF10A-AMPKAR2 cells were imaged in iGM as in *Figure 3A*, and treated at time 0 with 10 μM IA. Color scale bar on upper left indicates approximate AMPK index, with white representing low activity and orange/brown representing high activity. All supplemental movies were created using Nikon Elements software; ratiometric images of mTurquoise2 signal divided by YPet signal were constructed following background subtraction, and intensity modulation display was used to suppress spurious background values. Additional annotation was performed in ImageJ 2.0 (Fiji). Due to software limitations, the color scale bar was manually inserted using ImageJ and numerical values on the color bar do not correspond directly to AMPK index values shown in the main figures.
DOI: https://doi.org/10.7554/eLife.27293.016

**Video 2.** Fluctuating AMPK activity in response to oligomycin. MCF10A-AMPKAR2 cells were imaged in iGM as in *Figure 3B*, treated at time 0 with 4.5 μg/ml oligomycin. Color scale bar on upper right indicates AMPKAR signal, with white representing low activity and orange/brown representing high activity. Annotation was performed as for *Video 1*.
DOI: https://doi.org/10.7554/eLife.27293.017

working in reverse and pumping protons to maintain the mitochondrial electrochemical gradient; when proton flow is blocked by oligomycin (or potentially by high doses of CCCP), AMPK kinetics are determined by other processes, which we investigate below.

Using AMPKAR2/GMNN dual reporter cells, we examined cell fates in response to each metabolic inhibitor (*Video 4*). IA-treated cells rapidly ceased cell cycle progression (*Figure 3—figure supplement 5*) and underwent lysis at varying times 12–24 hr following treatment. In oligomycin-treated cells, cell cycle progression slowed but still led to normal mitoses, and viability was unaffected. While 1 μM CCCP induced greater average AMPK index than oligomycin, neither cell cycle progression nor cell viability were altered. Thus, cellular responses to metabolic perturbations do not correlate with overall magnitude of bioenergetic stress as measured by AMPKAR2. These differences in cell fate are likely to result from the modulation of distinct pathways by these agents, but it is nonetheless noteworthy that highly persistent dynamics are associated with more extreme changes in cell behavior, as this suggests that the kinetics of stress response may play a role in determining cell fate.

## Temporally coordinated oscillatory dynamics in bioenergetics and signaling upon inhibition of glycolysis

To understand why cells fail to reach stable adaptation under some conditions, we focused first on the rapid oscillations triggered by IA treatment (*Figure 4A*). The average period of these oscillations ranged from 18 min at 20–40 μM to 30 min at 5 μM (*Figure 4B*). For IA at 10 μM or greater, the percentage of cells displaying oscillations (defined as five or more successive pulses with a spacing of 1 hr or less) was >95%; this percentage fell to <60% at 5 μM IA, and no oscillation was detected at concentrations of 2.5 μM or less (*Figure 4C*). At intermediate concentrations (5–10 μM), IA-induced oscillations persisted for as many as 20 hr but typically ended with cell death (*Video 4*). Cells expressing Peredox or AKT-KTR treated with IA also exhibited oscillations in NADH index and Akt

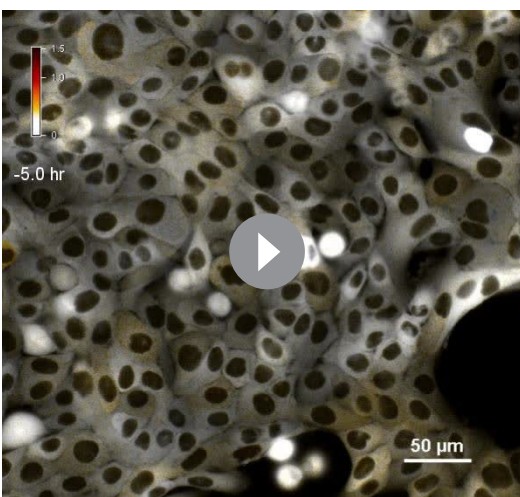

**Video 3.** Continuous AMPK activity in response to CCCP. MCF10A-AMPKAR2 cells were imaged in iGM as in **Figure 3C**, treated at time 0 with 1.6 µM CCCP. Color scale bar on upper right indicates AMPKAR signal, with white representing low activity and orange/brown representing high activity. Annotation was performed as for **Video 1**.
DOI: https://doi.org/10.7554/eLife.27293.018

we constructed a composite diagram of the relationship between the three parameters (**Figure 4H**). Thus, single-cell oscillations in PI3K/Akt activity, AMPK activity, and glycolytic NADH production were temporally coordinated, suggesting that feedback regulation tightly links these processes on the scale of minutes and leads to a persistent cycling of each pathway.

We hypothesized that IA-induced oscillations in AMPK activity and NADH/NAD$^+$ ratio resulted from oscillations in glycolytic flux, triggered by partial inhibition of GAPDH activity by 10 µM IA and the resulting feedback-driven increases in the entry of glucose into glycolysis. Analysis of glycolytic flux using $^{13}$C-labeled glucose confirmed that 10 µM IA reduced the appearance of labeled pyruvate and lactate by approximately 50%, while 50 µM IA reduced labeling by approximately 90% (**Figure 4—figure supplement 2**). Lactate secretion was reduced moderately by 10 µM IA and more strongly by 50 µM IA; together these measures indicate that glycolytic flux is reduced to an intermediate rate by 10 µM IA. We then compared the pulse scores for IA-treated cells in the presence of varying extracellular concentrations of glucose and pyruvate. In the absence of glucose, IA treatment failed to induce oscillations in AMPK or NADH index (**Figure 4—figure supplement 1B–C**). The incidence of oscillations, and the corresponding

index, respectively, with kinetics similar to those seen in the AMPK index (**Figure 4D and E**, **Videos 5** and **6**). Oscillations were not observed when using the Peredox control reporter Y98D (**Figure 4—figure supplement 1A**). To test whether the IA-induced oscillations in AMPK, Akt, and NADH indices were interrelated, we utilized AMPKAR2/AKT-KTR and Peredox/AKT-KTR dual reporter cells. AKT-KTR oscillations were tightly phase-locked with both AMPKAR2 and Peredox oscillations; each cycle of AKT-KTR response corresponded to one cycle of AMPKAR2 signal and one cycle of Peredox signal (**Figure 4F and G**). Phase locking was present in >90% of cells. In each cycle, peak signals of each reporter were phase-shifted relative to one another. In cells expressing Peredox and AKT-KTR, cycles initiated with a drop in NADH index that was followed approximately 0.25 cycles later by a drop in Akt index (**Figure 4F**). In cells expressing AMPKAR2 and AKT-KTR, the initial decrease in Akt index coincided with an increase in AMPK index, and peaks of Akt and AMPK index remained shifted by 0.5 cycles thereafter (**Figure 4G**). Based on these relative phase shifts,

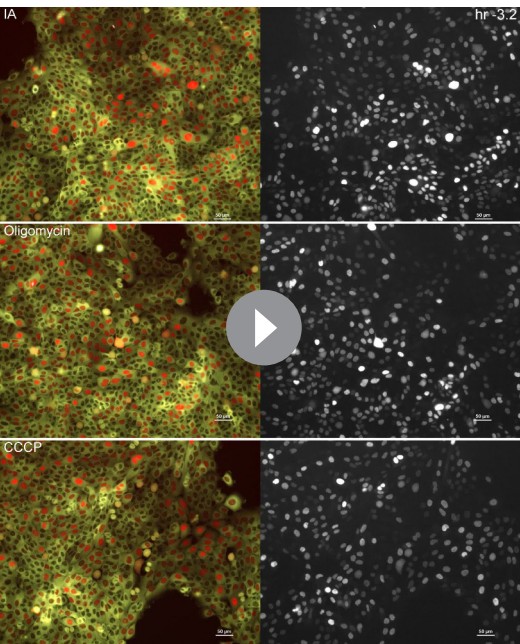

**Video 4.** Cell fates in response to metabolic perturbation. MCF10A-AMPKAR2-mCherryGMNN cells were imaged in iGM and treated with 10 µM IA, 4.5 µg/ml oligomycin, or 1.6 µM CCCP. The left column of images shows the YPet channel of AMPKAR as a cytosolic marker (yellow) and mCherryGMNN (red); the right column shows mCherryGMNN alone.
DOI: https://doi.org/10.7554/eLife.27293.019

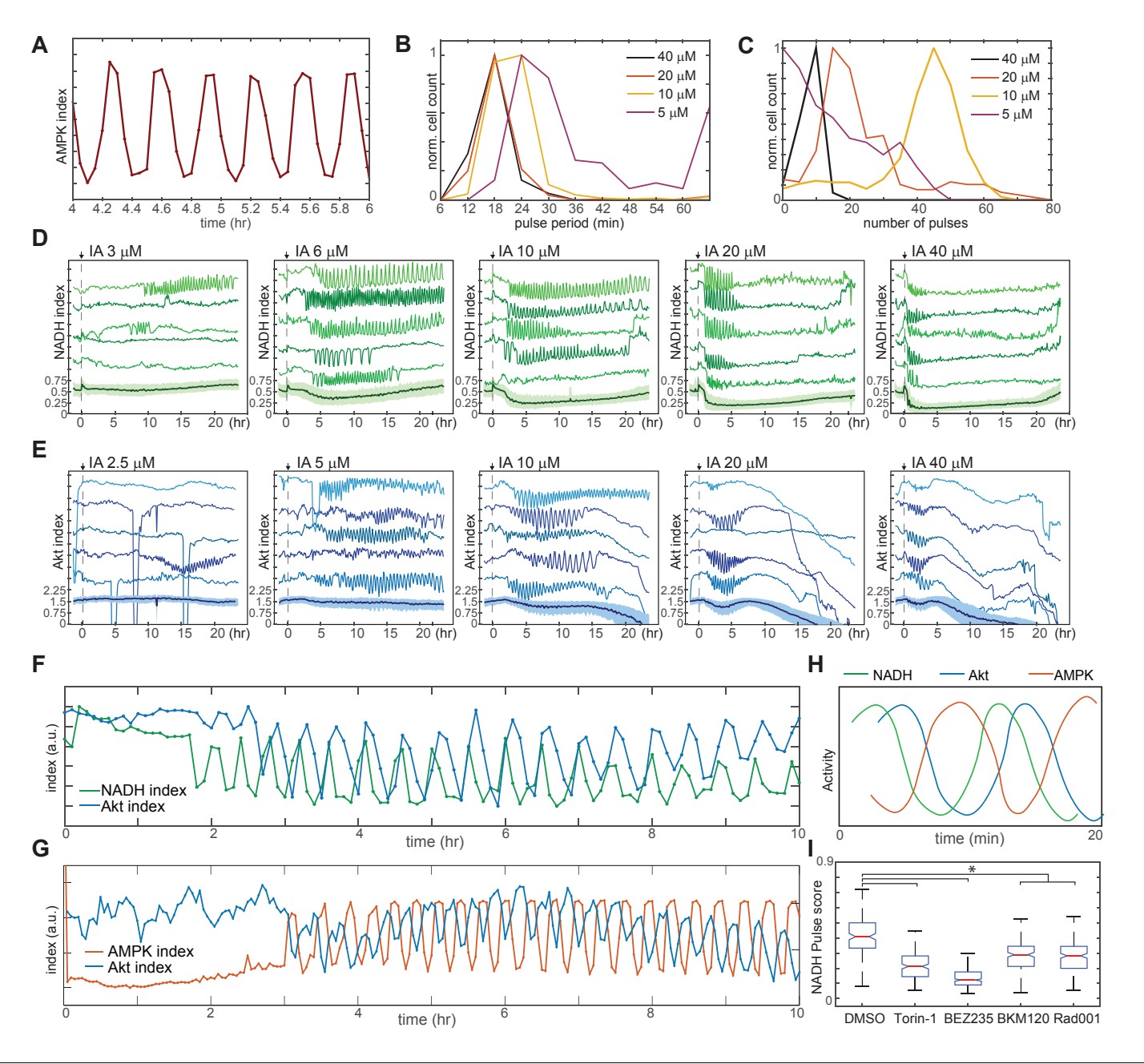

**Figure 4.** Linked oscillations in AMPK, Akt, and NADH indices triggered by inhibition of glycolysis. (**A**) Expanded view of a representative region of oscillatory AMPK index for cells cultured in iGM and treated with 10 µM IA. (**B**) Distribution of the average period of AMPK pulses for cells with five or more pulses, calculated as average peak-to-peak spacing following IA treatment as shown in *Figure 3A*. Distributions represent measurements of >500 cells under each condition. (**C**) Distribution of the number of pulses in AMPK index in each cell following IA treatment, measured as the largest series of detectable pulses spaced by 1 hr or less. (**D, E**) Single-cell measurements of NADH index (**D**) and Akt index (**E**) for the indicated concentrations of IA. Imaging medium was iGM lacking pyruvate, with IA added at time 0. $N = 4$, combined. (**F, G**) Simultaneous measurements of NADH and Akt indices (**F**) or AMPK and Akt indices (**G**) within the same cell. MCF10A cells expressing dual reporters were cultured and treated with 10 µM IA as in *Figures 4D* and *3A*, respectively. Cells shown were manually selected to best represent the phase relationship visible in the majority of cells. (**H**) Diagram of approximate phase relationship between NADH, AMPK, and Akt indices derived from the data collected in (**F**) and (**G**). (**I**) Pulse analysis of NADH index in IA-treated cells in response to inhibitors of PI3K/mTOR signaling. $N = 3–4$, representative. Statistical significance was determined as described in the Materials and methods section; all inhibitor conditions differed significantly from the DMSO condition, while Torin-1 (250 nM), BMK120 (250 nM), and Rad001 (20 nM) were significantly different from BEZ235 (250 nM).

DOI: https://doi.org/10.7554/eLife.27293.020

*Figure 4 continued on next page*

*Figure 4 continued*

The following figure supplements are available for figure 4:

**Figure supplement 1.** Analysis of conditions required for IA-induced metabolic oscillations.

DOI: https://doi.org/10.7554/eLife.27293.021

**Figure supplement 2.** Quantitative analysis of residual glycolytic flux in IA-treated MCF10A cells.

DOI: https://doi.org/10.7554/eLife.27293.022

**Figure supplement 3.** Effects of additional growth factor and fatty acid metabolism perturbations on IA-induced oscillations.

DOI: https://doi.org/10.7554/eLife.27293.023

**Figure supplement 4.** Proposed model of mechanisms underlying IA-induced oscillations.

DOI: https://doi.org/10.7554/eLife.27293.024

pulse score, increased with the extracellular glucose concentration in a dose-dependent manner, reaching a maximum at 4–5 mM, while the average period and amplitude remained essentially constant. In contrast, pyruvate alone, although capable of serving as an ATP source for MCF10A cells (*Figure 1—figure supplement 2A*) was unable to sustain IA-induced oscillations (*Figure 4—figure supplement 1D*). Pyruvate also had no effect on IA-induced AMPK index oscillations in the presence of glucose, although it rendered NADH index oscillations undetectable by lowering the resting NADH/NAD$^+$ ratio (*Figure 4—figure supplement 1D–E*). Together, these observations indicate that pyruvate does not fuel ATP production at a high enough rate to impact the rapid oscillatory changes during IA treatment. The data support the conclusion that these rapid oscillatory dynamics originate from changes in flux in glycolysis, with downstream metabolic processes playing little role.

Oscillations often arise in feedback systems in which there is a delay between induction of feedback and recovery of the feedback-controlled variables (*Glass et al., 1988*). The co-oscillation of Akt activity along with AMPK and NADH indices suggests the involvement of a complex feedback structure involving PI3K/Akt, AMPK, and also mTOR (*Figure 1—figure supplement 1*) (*Roberts et al., 2014*; *Yu et al., 2011*). Consistent with this idea, multiple inhibitors of mTOR and PI3K activity suppressed IA-induced NADH oscillations (*Figure 4I*). Suppression was most potent with BEZ235, which inhibits PI3K, mTORC1, and mTORC2 activity, and strong but somewhat less potent with Torin1, which inhibits both mTORC1 and mTORC2. Both Rad001, which inhibits mTORC1 alone, and BKM120, which inhibits only PI3K, had more limited but still significant ability to block oscillations. Withdrawal of insulin and EGF from the growth medium, which both reduces Akt and mTOR activity and glucose uptake (*Figure 2A and C* and *Worster et al., 2012*), also resulted in attenuation of IA-induced oscillations (*Figure 4—figure supplement 3A*), while inhibition of fatty acid oxidation, which could link pulses of AMPK activity to NADH production, reduced the persistence of AMPK index oscillations, but did not eliminate them (*Figure 4—figure supplement 3B*). Taken together, the observation that oscillations are most potently suppressed when multiple candidate feedback controllers are inhibited supports the concept of a multi-tiered feedback control system that does not depend critically on any single linkage. The phase relationship between the measurable variables in this system indicates the existence of delays between feedback activation and recovery of ATP and NADH and supports a model in which the slowed flux through glycolysis due to IA treatment triggers a cyclic series of feedback events that drive regular oscillations (*Figure 4—figure supplement 4*).

## Modulation of glucose metabolism controls stable adaptation to mitochondrial ATPase inhibition

Oligomycin-induced fluctuations in AMPK differed from IA-induced oscillations in their longer time scale (2–6 hr), in their irregular nature, and in that they were not tightly associated with coordinated changes in NADH and Akt indices (*Figure 5—figure supplement 1A–B*). To investigate the role of glycolysis in oligomycin-induced fluctuations, we examined the effect of glucose concentration in the absence of the alternate fuel sources glutamine and pyruvate (*Figure 5A*, *Video 7*). At 0 mM glucose, the baseline AMPK index was high, and oligomycin treatment led to a small increase in AMPKAR index with no subsequent adaptation, followed by cell death in 100% of cells within 12 hr. In 17 mM glucose (the baseline concentration for MCF10A media), oligomycin induced a rapid initial pulse of AMPK activity and subsequent adaptation, with >75% of cells returning to baseline AMPK index within 2 hr. Following this adaptation, cells displayed regular pulsatile dynamics in AMPK index, with

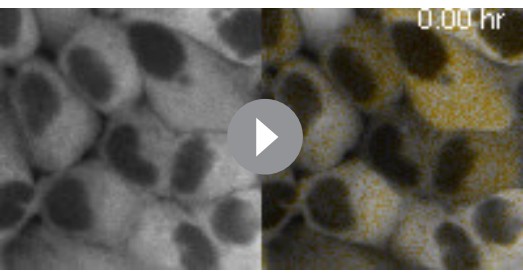

**Video 5.** AMPK and Akt activity oscillations in response to IA. MCF10A-AKT-KTR/AMPKAR2 cells were imaged in iGM2 as in **Figure 4E**, and were treated at time 0 with 10 μM IA. AMPK index is shown on right using a color scale as in **Video 1**. AKT-KTR signal is shown on left in grayscale.
DOI: https://doi.org/10.7554/eLife.27293.025

an average period of ~2.5 hr; the first two pulses of AMPK index were highly synchronous among cells, followed by gradual de-synchronization. As with the initial pulse, each burst of AMPK activity lasted 2–4 hr, suggesting that continuing oligomycin treatment induced ongoing bioenergetic challenges, which were, nevertheless, overcome by cells maintained at 17 mM glucose. At glucose concentrations of 3.4 mM and 1.7 mM, cells were unable to achieve full adaptation, with <25% and <10% of cells returning to baseline within 2 hr, and subsequent pulses in AMPK index were relatively dampened and prolonged. Thus, under high glucose levels, recovery of ATP levels occurs quickly and completely, but gives rise to recurring pulses of AMPK activity, suggesting that changes in the rate of ATP production by glucose metabolism via glycolysis generate these pulses.

We next examined how GF and nutrient availability affect these dynamics. In the presence of glutamine, treatment with insulin strongly suppressed oligomycin-induced pulses in AMPKAR index, relative to non-GF treated cells (**Figure 5B and C**). While EGF moderately increased the strength of pulses, co-treatment with both EGF and insulin led to suppression of pulses. This suppression was negated by co-treatment with Akt inhibitor, which strongly enhanced oligomycin-induced AMPKAR pulses (**Figure 5D and E**). Similarly, in the presence of insulin, moderate inhibition of glycolysis with a dose of IA too low to independently stimulate oscillations resulted in amplification of oligomycin-induced AMPK pulses (**Figure 5—figure supplement 1C**). Glutamine was required for the suppression of oscillations by insulin (**Figure 5—figure supplement 1D**), but regardless of the presence of insulin, cells cultured in the presence of glutamine without glucose failed to recover their AMPK index and died within 12 hr (**Figure 5—figure supplement 1E**), suggesting that glutamine may play a role in supporting glucose metabolism, but cannot alone provide sufficient ATP in the absence of oxidative phosphorylation. Pyruvate had no effect on the kinetics of oligomycin response (**Figure 5—figure supplement 1F**). Altogether, these data suggest a model in which increased flux through glycolysis stimulated by insulin is able to attenuate recurrent ATP shortages during continued inhibition of oxidative phosphorylation (**Figure 5—figure supplement 2**).

## Akt-stimulated glucose uptake is required for bioenergetic stability in proliferation

Finally, we returned to the question of how GF stimulation and proliferation impact bioenergetic stability even in the absence of overt metabolic perturbations. We quantified the AMPK pulse scores for GF-stimulated MCF10A cells; we found that while AMPK index fluctuations under these conditions were less pronounced than in IA- or oligomycin-stressed cells, significantly more fluctuations occurred in non-GF- or EGF-treated cells relative to cells treated with insulin or a combination of insulin and EGF (**Figure 6A and B**). To find the basis for these differences, we first examined AMPK index fluctuation over the cell cycle. AMPK activity was more pulsatile in G0/G1 cells relative to cells in S/G2/M phases of the cell cycle (**Figure 6C**). However, this

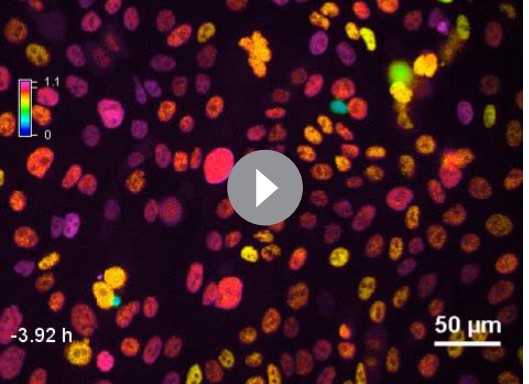

**Video 6.** NADH oscillations detected by Peredox in response to IA. MCF10A-Peredox cells were imaged in iGM2 as in **Figure 4D** and were treated at time 0 with 10 μM IA. Color scale bar at upper left indicates NADH index, with blue representing low values and green, yellow, red, and violet indicating successively higher values.
DOI: https://doi.org/10.7554/eLife.27293.026

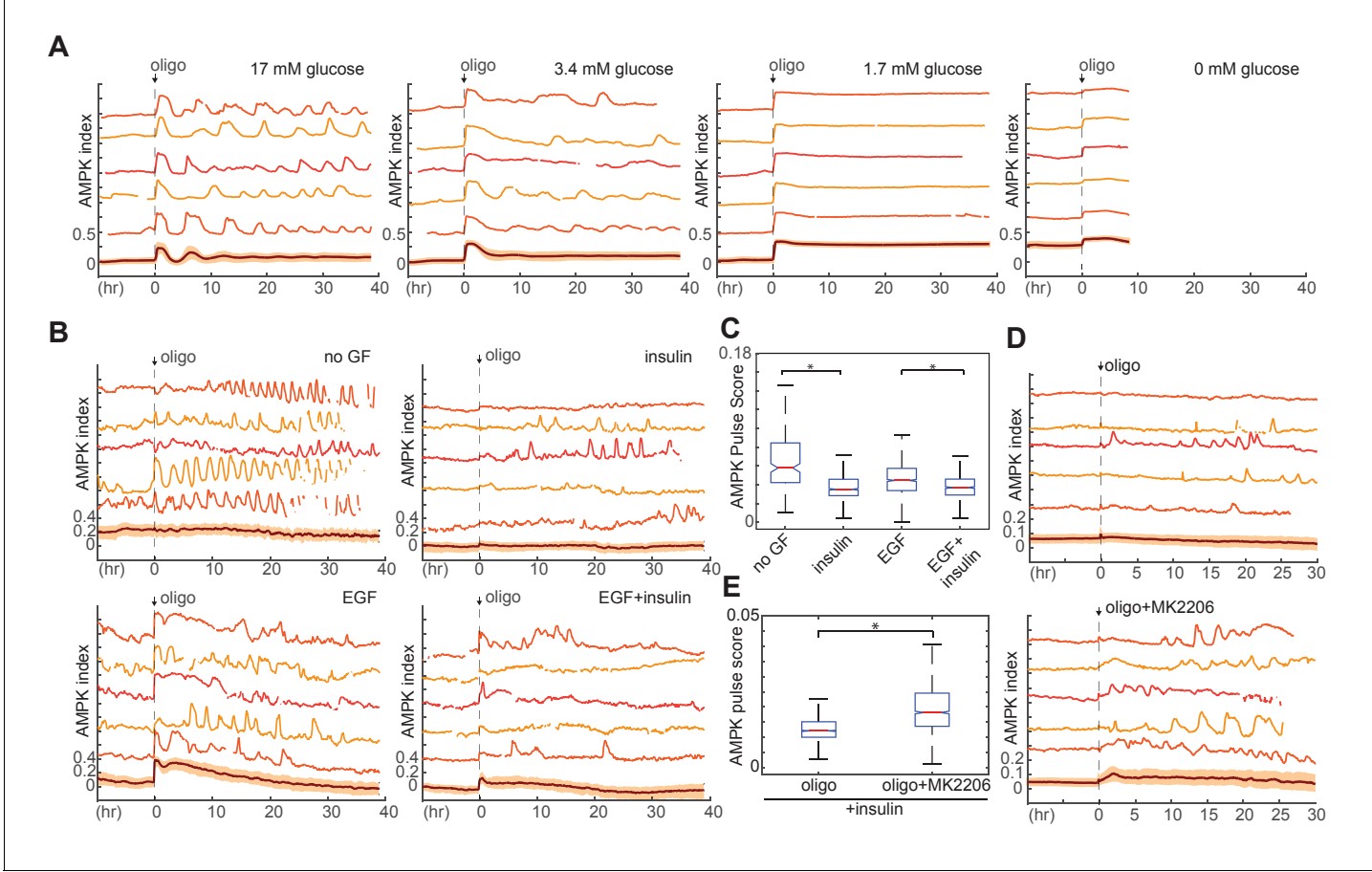

**Figure 5.** Requirement of glucose and glycolytic metabolism for AMPK fluctuations. (**A**) Single-cell measurements of AMPK index in MCF10A cells in response to oligomycin at different glucose concentrations. Prior to imaging, cells were placed in iGM lacking pyruvate and glutamine at different glucose concentrations, and were treated at time 0 with 1.8 µg/ml oligomycin. Measurements under 0 mM glucose are truncated due to cell death that began approximately 5 hr following oligomycin treatment. N = 5, representative. (**B**) Single-cell measurements of AMPK index in the presence of EGF, insulin, or both, with exposure to 1.8 µg/ml oligomycin at time 0. Prior to imaging, cells were placed in iGM lacking pyruvate, with 2.5 mM glutamine, 3.4 mM glucose, and 20 ng/ml EGF or 10 µg/ml insulin. (**C**) Quantification of pulse score for each condition shown in (**B**). N = 5, representative. Statistical significance between no GF and insulin conditions, and between EGF and EGF +insulin conditions, was determined as described in the Materials and methods section. (**D**) Single-cell measurements of AMPK index in MCF10A cells treated with 1.8 µg/ml oligomycin, with or without 250 nM Akt inhibitor (MK2206) at time 0. Prior to imaging, cells were placed in iGM lacking pyruvate and EGF and containing 2.5 mM glutamine, 3.4 mM glucose, and 10 µg/ml insulin. (**E**) Quantification of pulse score for each condition shown in (**D**). N = 3, representative. Statistical significance was determined as described in the Materials and methods section.

DOI: https://doi.org/10.7554/eLife.27293.027

The following figure supplements are available for figure 5:

**Figure supplement 1.** Additional measurements of metabolic responses to oligomycin.
DOI: https://doi.org/10.7554/eLife.27293.028
**Figure supplement 2.** Model for AMPK oscillations induced by oligomycin.
DOI: https://doi.org/10.7554/eLife.27293.029

moderate difference between cell cycle phases does not explain the overall effect of GFs on AMPK kinetics: compared to EGF-treated cells, insulin-treated cells are more likely to be in G0/G1 but have a lower probability of AMPK index fluctuations. We therefore next investigated the involvement of Akt by simultaneously monitoring both Akt and AMPK in dual reporter MCF10A-AMPKAR2/AKT-KTR cells. Analysis of fluctuations in both reporters on a cell-by-cell basis revealed a high frequency of inverse events, with a pulse in AMPK index mirrored by a decrease in Akt index (*Figure 6D*). Cross-correlation analysis indicated that such inverse events were highly overrepresented in the

**Video 7.** AMPK activity dynamics as a function of glucose level. MCF10A-AMPKAR2 cells were imaged in iGM modified to lack glutamine and pyruvate and to contain the indicated concentrations of glucose. Oligomycin (9 µg /ml) was added at time 0. Annotation was performed as for *Video 1*.
DOI: https://doi.org/10.7554/eLife.27293.030

population relative to their expected occurrences at random (*Figure 6E*), suggesting that AMPK fluctuations may result at least in part from rises and falls in Akt activity and the associated rate of glucose uptake.

We further tested the role of the PI3K/Akt pathway in bioenergetic stress using pharmacological inhibitors of this pathway. In cells growing in the presence of either insulin alone or a combination of insulin and EGF, treatment with either PI3K inhibitor (BKM120) or Akt inhibitor (MK2206) increased the AMPK pulse score (*Figure 7A and B*). Similarly, NADH index fluctuations were increased with Akt, PI3K, mTOR, or dual PI3K/mTOR inhibitors (*Figure 7C and D*). We conclude that PI3K/Akt signaling plays an active role in suppressing fluctuations in bioenergetic stress in cells at both high and low proliferation rates.

## Discussion

The kinetics of metabolic homeostasis at the organismal level are known in detail (for example, blood glucose clearance rates are well characterized and widely used as diagnostics), but metabolic homeostasis at the single-cell level has remained largely unexplored. Given the extensive interconnections between metabolic regulation and GF signaling at the level of Akt, mTOR, and AMPK, intricate metabolic dynamics in single cell physiology have long been postulated (*Plas and Thompson, 2005*) but never examined in detail. Here, using a panel of fluorescent biosensors for key metabolic regulators, we show that individual cells experience frequent deviations in bioenergetic and signaling parameters, both during proliferation and in response to metabolic challenges. As fluctuations in ATP and NADH availability can influence functions such as DNA synthesis and gene expression, understanding these metabolic dynamics, rather than simply average or baseline concentrations, will be crucial in developing an integrated model for the control of cellular metabolism and growth.

Our results point to a central role for insulin/PI3K/Akt regulation of glycolysis in modulating cellular metabolic stability. Despite its relative inefficiency in ATP yield per molecule of glucose, glycolysis can, at least under certain conditions, produce ATP at a faster rate than oxidative phosphorylation if sufficient glucose is available; this ability is best documented in muscle cells during anaerobic activity but could conceivably extend to other situations such as hypoxic cells within a tumor (*Liberti and Locasale, 2016*). The data presented here suggest that the regulation of glycolysis by glucose availability and insulin/PI3K signaling can play both positive and negative roles in bioenergetic stability. For example, insulin enhances the occurrence of IA-induced oscillations (*Figure 4I*), but has a suppressive role for oligomycin-induced oscillations (*Figure 5C*). In our models, this difference is consistent with the configuration of the network in each case. In the case of IA treatment, where a bottleneck is imposed between upper and lower glycolysis, higher glucose input stimulated by insulin would, by mass action, force increased flux through this bottleneck. In a situation where delayed feedback drives oscillations (See *Figure 4—figure supplement 4*), this increased flux would amplify the resulting oscillations. In contrast, during treatment with oligomycin, increasing the flux through glycolysis by treatment with insulin effectively bypasses the imposed blockade on TCA production of ATP, dampening the effects of negative feedback (*Figure 5—figure supplement 2*). Thus, the ability of glycolytic flux to adapt quickly may provide the ability to quickly restore ATP levels when they fall, facilitating energetic stability under some conditions. In other circumstances, the rapid changes in ATP and other metabolites made possible by glycolysis may expose the cell to oscillatory behavior, as slower regulatory processes attempt to catch up.

Because ATP plays a central role in providing energy for many essential cellular processes, even short lapses in availability can potentially compromise cellular function and viability; it is likely that evolution has selected for feedback kinetics that rapidly reverse any decrease in ATP to prevent levels from falling dangerously low. Consistent with this idea, we find that cells provided with different fuel sources (glucose, glutamine, and pyruvate) are able to adapt and maintain steady levels of

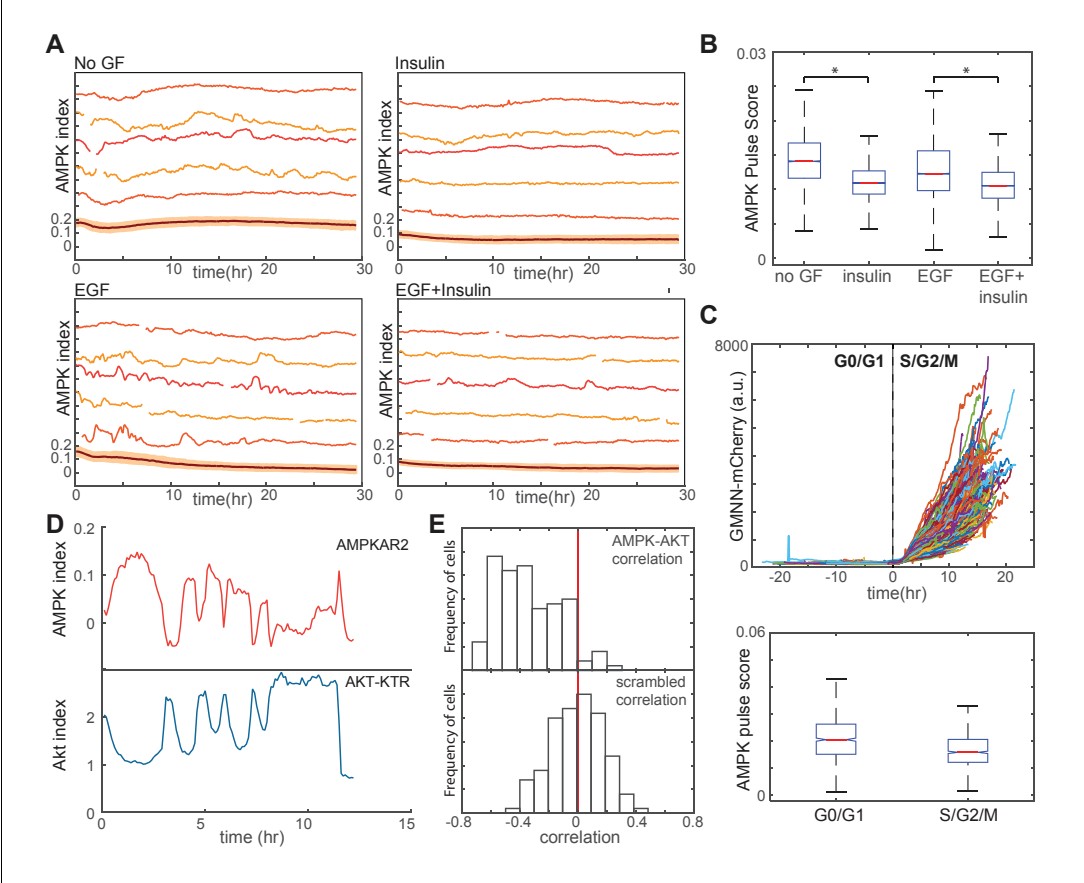

**Figure 6.** Linkage of AMPK and Akt activity fluctuations in the absence of chemical metabolic stresses. (A,B) Single-cell measurements of AMPK index in MCF10A cells in the presence of EGF (20 ng/ml), insulin (10 μg/ml), or both. Prior to imaging, cells were placed in iGM lacking pyruvate and containing the indicated GFs. Quantification of pulse score for each condition is shown in (B). N = 4, representative. Statistical significance between no GF and insulin conditions, and between EGF and EGF +insulin conditions, was determined as described in the Materials and methods section. (C) Quantification of pulsatile behavior in G0/G1 relative to S/G2. Live-cell measurements of Geminin-mCherry and AMPK index were made in individual cells grown in the presence of EGF; cells were aligned by the time of G1-to-S transition (top). For each cell, we calculated the pulse scores during a 10 hr window preceding the induction of GMNN-mCherry to a 10 hr window immediately following induction (bottom). (D) Single-cell measurements of AMPK index (AMPKAR2) and Akt index (AKT-KTR) in the same cell; culture medium was iGM lacking insulin and pyruvate. The cell shown was manually selected to display a prominent example of the anti-correlation trend visible in the population analysis in (E). (E) Correlation analysis of AMPK and Akt indices. The time-dependent correlation between measured AMPK and Akt was quantified for each cell, and the distribution of individual correlation values is shown. A histogram centered around 0 indicates a population in which there is no systematic trend in the correlation, while skew to the left or right indicates a negative or positive trend in correlation, respectively. The bottom panel shows distributions of correlation values between Akt and AMPK for randomly chosen cells as a negative control.

DOI: https://doi.org/10.7554/eLife.27293.031

AMPK activity with few fluctuations, albeit at different set points that depend on the fuel source (*Figure 1—figure supplement 2A,B*). However, optimization for such rapid and efficient adaptation comes with the potential that for certain conditions stable adaptation cannot be achieved, and unstable (e.g. oscillatory) responses result (*Chandra et al., 2011*). In terms of nonlinear dynamics, such responses occur in limited regions of parameter space near unstable fixed points. Accordingly, we find that epithelial cells can be forced into persistent oscillatory behavior within certain intermediate conditions. For example, IA-induced oscillations are most persistent at intermediate doses of 10–20 μM; at lower doses, cells successfully adapt after brief cycling, whereas at higher doses, cells simply fail to adapt and remain at a high level of AMPK activity and low NADH. The conditions where we observe unstable behavior – including inhibition of lower glycolysis or mitochondrial ATP production, or culture in the complete absence of insulin-stimulated glucose uptake – likely represent situations that are unusual under normal physiological function. However, such conditions may

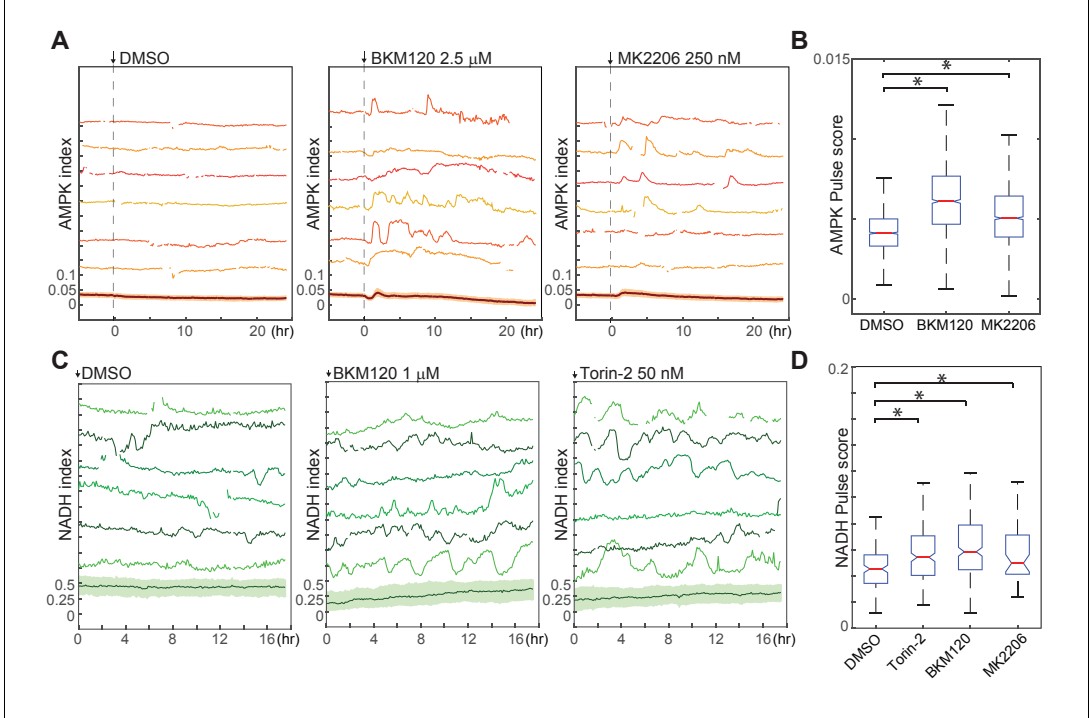

**Figure 7.** Suppression of spontaneous bioenergetic fluctuations by Akt/PI3K signaling. (A,B) Single-cell measurements (A) and pulse analysis of AMPK index (B) in MCF10A cells imaged in iGM and treated with BKM-120 or MK2206 at time 0. N = 4, representative. Statistical significance was determined as described in the Materials and methods section; both inhibitor conditions differed significantly from the DMSO condition with p<0.01. (C, D) Single-cell measurements (C) and pulse analysis (D) of NADH index in MCF10A cells expressing Peredox and cultured in iGM2 with the indicated inhibitors. Median pulse scores are shown in green above each plot. N = 2, combined. Because the Peredox signal showed a higher degree of frame-to-frame variance that is not likely to be biologically relevant, the 'smooth' parameter was set to six for this analysis to prevent identification of spurious pulses.
DOI: https://doi.org/10.7554/eLife.27293.032

occur under pathological circumstances, including mutations of metabolic enzyme genes or pharmacological or toxic compounds that impair metabolic function and may be important for the understanding of tissue function in these cases. In the epithelial cells examined here, metabolic oscillations coincide with impairment of cellular function ranging from slowed proliferation to cell death, suggesting that the oscillations represent a deleterious side effect of feedback control. On the other hand, it has also been proposed that sustained glycolytic oscillations could play an important role in fine-tuning certain cellular functions, such as insulin secretion in pancreatic beta cells (*Goodner et al., 1977*).

Previous studies of glycolytic oscillations have measured periods ranging from several seconds to 20 min (*Chou et al., 1992*; *O'Rourke et al., 1994*; *Tornheim and Lowenstein, 1973*; *Yang et al., 2008*). Oscillations have been recapitulated in isolated extracts of both yeast and mammalian myocytes, indicating that the core glycolytic enzymes alone are sufficient, with the allosteric regulation of PFK playing a central role. Our observations differ from these studies in the longer period of the oscillations (20–30 min in the case of glycolytic inhibition by IA; 3–6 hr in the case of mitochondrial inhibition by oligomycin), as well as in implicating a role for Akt and AMPK in oscillations. We speculate that in the epithelial cells examined here, a core glycolytic oscillator becomes entrained through feedback connections to these additional regulatory pathways that are central to growth and homeostasis in this cell type. Our data also suggest that metabolic oscillations may be a wider phenomenon than previously thought, as we demonstrate their occurrence in cells not typically considered highly metabolically active, and also find that they may occur with heterogeneous phasing that makes oscillations impossible to detect without single cell methods. The tools developed here will be of use in detecting and analyzing similar oscillations in other cell types and conditions.

Given that over 90% of human tumors arise in epithelial tissue and that abnormal cell proliferation underlies tumorigenesis, understanding metabolic requirements for proliferating epithelial cells can have profound implications in oncology research. Our findings offer a potential explanation for the metabolic advantage conferred by aerobic glycolysis in tumors and proliferating cells. Existing hypotheses for why aerobic glycolysis is common in proliferating cells include rapid ATP generation by glycolysis (though less efficient than oxidative phosphorylation), as well as increased fluxes to glycolytic intermediates for biosynthesis (*Sullivan et al., 2015*). We find that in the presence of EGF, where ATP and NADH are low but proliferative rate is high, cells display increased bioenergetic instability and sensitivity to inhibition of oxidative respiration, which can be reversed by insulin-mediated activation of Akt. In the context of tumor microenvironments with fluctuating nutrient and oxygen supply, such instability is likely deleterious and may create selective pressure for genetic alterations to enhance glycolysis, such as activating mutations in the PI3K/Akt pathway, which are among the most frequent mutations across all cancer types. Investigating the effects of oncogenic mutations on metabolic stability may thus be important in developing therapies that target the altered metabolism of tumor cells.

# Materials and methods

## Key resources table

| Reagent type (species) or resource | Designation | Source or reference | Identifiers | Additional information |
|---|---|---|---|---|
| Cell line (*H. sapiens*) | MCF10A | Brugge Lab, Harvard Medical School | RRID: CVCL_0598 | Clone 5E (*Janes et al., 2010*) used for all experiments |
| Cell line (*H. sapiens*) | MCF12A | ATCC | RRID: CVCL_3744 | |
| Cell line (*H. sapiens*) | 184A1 | ATCC | RRID: CVCL_3040 | |
| Cell line (*H. sapiens*) | MCF10A-AMPKAR2 | This paper | N/A | Available on request |
| Cell line (*H. sapiens*) | MCF10A-AMPKAR2-GMNN | This paper | N/A | Available on request |
| Cell line (*H. sapiens*) | MCF10A-AMPKAR2-AKTKTR | This paper | N/A | Available on request |
| Cell line (*H. sapiens*) | MCF10A-Peredox | This paper | N/A | Available on request |
| Cell line (*H. sapiens*) | MCF10A-Peredox-AKTKTR | This paper | N/A | Available on request |
| Peptide, recombinant protein | Epidermal growth factor | Peprotech | Cat#AF-100–15 | |
| Peptide, recombinant protein | Insulin | Sigma-Aldrich | Cat#I9278 | |
| Antibody | Anti-pACC | Cell Signaling Technology | Cat#11818 RRID:AB_2687505 | |
| Chemical compound, drug | Sodium iodoacetate | Sigma-Aldrich | Cat#I2512 | |
| Chemical compound, drug | Carbonyl cyanide 3-chlorophenylhydrazone | Sigma-Aldrich | Cat#C2759 | |
| Chemical compound, drug | Oligomycin A | Sigma-Aldrich | Cat#75351 | |
| Chemical compound, drug | BKM120 | Axon Medchem | Cat#1797 | |
| Chemical compound, drug | MK2206 | Selleck Biochemicals | Cat#S1078 | |
| Chemical compound, drug | BEZ235 | Axon Medchem | Cat#1281 | |
| Chemical compound, drug | Torin-1 | Tocris Biosciences | Cat#4247 | |
| Chemical compound, drug | Torin-2 | Selleck Biochemicals | Cat#S2817 | |
| Chemical compound, drug | Rad001 | Selleck Biochemicals | Cat#S1120 | |
| Transfected construct (synthetic) | pPBJ-puro-AMPKAR2-nes | This paper | N/A | Available on request |
| Transfected construct (synthetic) | pMSCV-puro-AKTKTR-mCherry | This paper | N/A | Available on request |
| Transfected construct (synthetic) | pMSCV-puro-Peredox-nls-mCitrine | This paper | N/A | Available on request |

*Continued on next page*

*Continued*

| Reagent type (species) or resource | Designation | Source or reference | Identifiers | Additional information |
|---|---|---|---|---|
| Transfected construct (synthetic) | pMSCV-puro-GMNN-mCherry | *Albeck et al. (2013)* | N/A | Available on request |
| Software, algorithm | MATLAB | Mathworks | RRID: SCR_001622 | |

## Reporter construction

Peredox-mCitrine-NLS was constructed by replacing mCherry with mCitrine (*Shaner et al., 2005*) in pMSCV-Peredox-mCherry-NLS (*Hung et al., 2011*). The negative control with abrogated NADH binding was constructed by introducing the mutations Y98D in both subunits of Rex and I189F in the first subunit of Rex in pMSCV-Peredox-mCherry-NLS. AKT-KTR was constructed by fusing the N-terminal domain (amino acid residues 1–400) of human FOXO3a DNA binding mutant H212R (*Tran et al., 2002*; from Addgene) with a C-terminal mCherry in the retroviral pMSCV vector. AMP-KAR2-EV was constructed by modifying the linker between the CFP and YFP in AMPKAR (*Tsou et al., 2011*) with an expanded EEVEE linker (*Komatsu et al., 2011*), and replacing the CFP and YFP fluorophores with mTurquoise2 and YPet, respectively; a PiggyBAC transposase-mediated delivery system (*Yusa et al., 2011*) was used to minimize recombination between CFP and YFP.

## Reagents

Reagents were from Sigma (St. Louis, MO) unless noted. Iodoacetate, lactate, pyruvate, and cyclo-heximide stocks were prepared in water. Rotenone, oligomycin A, and AICAR were dissolved in DMSO. BEZ235 (Axon Medchem, Reston, VA), Torin-1 (Tocris Bioscience, Bristol, UK), Torin-2 (Selleck, Houston, TX), BKM120 (Axon Medchem), GDC0941 (Axon Medchem), Gefitinib (Axon Medchem), MK2206 (Selleck), PD 0325901 (Sigma and Selleck), and Rad001 (SU2C PI3K Dream Team Mouse Pharmacy, which obtains compounds from Shanghai Haoyuan Chemexpress; [*Elkabets et al., 2013*]) were dissolved in DMSO. For GF titration, EGF (Peprotech, Rocky Hill, NJ) and insulin (Sigma) were diluted in PBS and added at indicated concentrations.

## Cell culture and media

Human mammary epithelial MCF10A cells clone 5E (*Janes et al., 2010*) were obtained from the Brugge lab and cultured as previously described (*Debnath et al., 2003*). The MCF10A full growth medium consisted of Dulbecco's modified Eagle's medium (DMEM)/F-12 (#11330, Life Technologies, Carlsbad, CA), supplemented with 5% horse serum (Life Technologies), EGF (20 ng/ml), insulin (10 µg/ml), hydrocortisone (0.5 µg/ml), cholera toxin (100 ng/ml), and penicillin (50 U/ml) and streptomycin (50 µg/ml). MCF12A and 184A1 cell lines were obtained directly from the ATCC (Manassas, VA) and were cultured under the same conditions as MCF10A. Because primary stocks from the original clonal derivation (MCF10A-5E) or from the ATCC (MCF12A and 184A1) were used in all experiments, further authentication was unnecessary. Cell lines stably expressing biosensors were generated by retroviral or lentiviral infection, or by transfection with the PiggyBac transposase system, (*Yusa et al., 2011*), followed by puromycin (1–2 µg/ml) selection and expansion of single clones. For each reporter, we isolated multiple stable clones with homogenous expression; all data reported in this study reflect representative behaviors that were highly consistent across all clones of each reporter (a minimum of three clones in each case). All reporter lines were confirmed to be mycoplasma-negative by PCR; results were validated by third-party testing of selected lines (ATCC).

For microscopy, we used a custom formulation with minimal background fluorescence, termed imaging-modified growth medium (iGM), which consists of DMEM/F12 lacking riboflavin, folic acid, and phenol red (Life Technologies). Lacking these three components had no effect on reporter kinetics, as indicated by experiments performed in normal growth medium lacking only phenol red (not shown), but allowed for more accurate quantification of reporter signals. iGM contained all supplements used in MCF10A culture medium (above), but with horse serum replaced by 0.1% (w/v) bovine serum albumin. In order to lower the amounts of extracellular pyruvate and facilitate measurements of cytosolic NADH-NAD$^+$ redox (*Hung and Yellen, 2014*) in some experiments, we used an alternate

imaging medium formulation, termed iGM2, which consists of 95% DMEM (Life Technologies #31053) and 5% DMEM/F12 (Life Technologies #11039), supplemented with 0.3% bovine serum albumin, EGF, insulin, hydrocortisone, cholera toxin, and pen-strep as above. For Peredox calibration using lactate and pyruvate, DMEM (Life Technologies #31053) with indicated lactate and pyruvate concentrations were made; cells were washed one to two times with the media prior to imaging. For GF titration, cells were placed in EGF/insulin-deficient medium for 2 days prior to imaging with appropriate concentrations of EGF and insulin. For glucose titration, DMEM (Life Technologies #A14430) with indicated glucose concentrations were prepared (with residual F12 supplementation of 0.8% in media with 0.03 mM to 23 mM glucose); cells were washed two to three times with the respective media prior to imaging.

For labeling experiments, cells were incubated with 0 µM, 10 µM, or 50 µM IA for 6 hr after which the culture medium was replaced with medium containing [U-13C6]glucose for 10 min prior to metabolite extraction.

## Live-cell fluorescence microscopy

Time-lapse wide-field microscopy was performed as previously described (*Hung et al., 2011*; *Albeck et al., 2013*). Briefly, 1000–2500 cells were seeded 2–4 days prior in glass-bottom 24-well (MatTek) or 96-well plates (MGB096-1-2-LG-L; Brooks Life Sciences, Chelmsford, MA), with well bottom pretreated with type I collagen (BD Biosciences, San Jose, CA) to promote cell adherence. For experiments with drug addition, cells were placed in imaging medium until the addition of drug diluted as a 5x-20x concentrated spike. Cells were maintained in 95% air and 5% $CO_2$ at 37°C in an environmental chamber. Images were collected with a Nikon (Tokyo, Japan) 20×/0.75 NA Plan Apo objective on a Nikon Eclipse Ti inverted microscope, equipped with a Lumencor SOLA or a Nikon Intensilight C-HGFI light source. Fluorescence filters were from Chroma (Bellows Falls, VT): T-Sapphire (89000 ET Sedat Quad; or ET405/20x, T425LPXR, and ET525/50 m), YFP (89002 ET ECFP/EYFP; or 41028), and RFP (49008 ET mCherry; or 41043 HcRed). With a Hamamatsu Photonics (Hamamatsu, Japan) ORCA-ER or ORCA-AG cooled CCD camera or an Andor Zyla 5.5 scMOS camera, images were acquired every 3–8 min with 2 × 2 binning and an exposure time of 200–225 ms for T-Sapphire, 70–225 ms for YFP, 70–225 ms for RFP.

## Immunofluorescence microscopy

Cells were plated in glass-bottom plates and treated as for live-cell microscopy, as described above. At indicated times during the experiment, medium was removed and cells were fixed with 4% paraformaldehyde for 15 min, followed by permeabilization with 100% methanol. Cells were then washed, blocked with Odyssey Blocking Buffer (Li-Cor, Lincoln, NE), and incubated with anti-pACC (Cell Signaling , Danvers, MA #11818) diluted 1:125 in blocking buffer, overnight at 4°C. Secondary staining was performed with Alexa647-conjugated anti-rabbit (Life Technologies), followed by DNA staining with Hoechst-33342. Plates were imaged as described for live-cell microscopy, using Chroma filter sets for Hoechst (49000 ET) and Alexa 647 (49006 ET).

## Image processing

Single-cell traces were generated using the automated software DCellIQ (*Li et al., 2010*), followed by manual verification using a custom MATLAB program (MathWorks, Natick, MA) to correct tracking errors, or using a custom MATLAB image processing pipeline (*Sparta et al., 2015*) using global optimization of cell tracks (*Jaqaman et al., 2008*). After background subtraction, DCellIQ or the MATLAB pipeline were used for image segmentation and tracking to determine nuclear masks based either on nuclear-localized Peredox-mCitrine or on the absence of YPet nuclear fluorescence of cytoplasmic-localized AMPKAR2. For a subset of the data, we additionally verified the automated tracking results manually. After cell tracking with the YFP images, the coordinates were applied to the other fluorescent channels. The nuclear masks were eroded by 1 µm to ensure the exclusion of cytoplasmic pixels; the nuclear T-Sapphire, CFP, YFP, and RFP signals were calculated as the mean pixel values within the nuclear masks in the respective images. The cytoplasmic CFP, YFP, and RFP signals were calculated as the mean pixel value within a cytoplasmic 'donut' mask, which consisted of an outer rim 3–4 µm from the nuclear mask and the inner rim as the perimeter of the eroded nuclear mask or 2–3 µm from the original nuclear mask. NADH index was calculated as a ratio of the

background-subtracted nuclear T-Sapphire to YFP signal. Akt index was calculated as a ratio of the background-subtracted cytoplasmic RFP to nuclear RFP signal. AMPK index was calculated as the ratio of the background subtracted cytoplasmic CFP to YFP ratio; because this ratio is linearly related to the fraction of unphosphorylated reporter molecules (*Birtwistle et al., 2011*), this signal was inverted by multiplying it by $-1$ and adding a positive value to set the lowest AMPK index in each experiment to approximately 0.

## Analysis and statistics of kinetics in reporter signals

A custom MATLAB algorithm was designed to identify peaks in the time-lapse signal of AMPK or NADH index within each cell. The AMPK or NADH index was first smoothed to remove spurious noise. Peaks and associated valleys in the index were identified by setting two local cutoff values, based on maximum and minimum values of the data within a sliding time window (typically 40 min). A peak was detected if both cutoff values were crossed by a rise and subsequent fall in the index. To define a 'pulse score' for each cell, the amplitudes (difference between baseline and peak value) for all detected peaks were summed and normalized by the length of time the index was recorded. The pulse score for each cell thus increases with both the frequency and amplitude of peaks; examples of peak detection and corresponding scores are shown in *Figure 3—figure supplement 1*. Typically, >500 individual cell recordings were scored for each condition and plotted using a box plot, with the median shown as a red line, 25th-75th percentile (the interquartile range) as the bottom and top of the box, respectively. Whiskers indicate the range of values falling within 1.5 times the interquartile range outside of the interquartile range; outliers beyond this range are not shown. Statistical comparisons between samples are displayed using notches on the box plots; two samples in which notches do not overlap differ in their medians at the 5% significance level.

## Metabolite extraction and derivatization

[U-13C6] glucose-containing medium was aspirated and cells were washed once with ice-cold saline, after which 500 µl of $-20°C$ methanol was added to each well to quench metabolism. Next, 300 µl of ice cold ultrapure water containing 2 µg norvaline as an internal standard was added to a single well and cells were scraped into the methanol-water mixture and transferred to a microcentrifuge tube. This process was repeated for each well of the plate, after which 600 µl $-20°C$ chloroform was added to each tube and all tubes were vortexed at 4°C for 10 min. The metabolite extracts were then centrifuged at 4°C at 17,200 x g for 10 min, and the resulting upper (polar metabolite) phases of the extracts were transferred to new tubes. These polar metabolite extracts were then dried in a centrifugal evaporator and stored at $-80°C$ until derivatization for GC-MS analysis.

Metabolite extracts were derivatized by a two-step process. First, 15 µl of methoxyamine in pyridine (MOX Reagent, ThermoFisher) was added and the extracts were incubated at 40°C for 1.5 hr. Second, 20 µl N-(tert-butyldimethylsilyl)-N- methyl-trifluoroacetamide with 1% tert-butyldimethyl-chlorosilane (TBDMS) (Sigma) was added and samples were incubated at 60°C for 1 hr. The reaction mixtures were quickly vortexed and centrifuged, and supernatants were transferred to GC-MS vials for analysis.

## GC-MS analysis

Derivatized metabolite samples were analyzed on a 6890N GC with a DB-35ms Ultra Inert capillary column coupled to a 5975B Inert XL MS (Agilent, Santa Clara, CA). The flow rate of the helium carrier gas (Airgas, Radnor, PA) was maintained at 1 ml/min. The inlet temperature was held at 270°C. Injection volume was 2 µl and was not split. Selected ion monitoring mode was used to detect measured ions, with monitored m/z parameters identical to previously published values (*Ahn and Antoniewicz, 2011*). The instrument was operated in electron ionization mode with an energy of 70 eV. The GC oven was first held at 100°C for 3 min, then ramped at 2.5 °C/min to 300°C. Raw abundance data were integrated to give mass isotopomer distributions and subsequently corrected for natural abundance (*Fernandez et al., 1996*) using an in-house software operating in Matlab (MathWorks). To quantify relative pool sizes, metabolite abundances were normalized to both the norvaline internal standard abundance and protein content of proxy wells.

## Lactate secretion

Medium was harvested after 6 hr of treatment with 0 μM, 10 μM, or 50 μM IA and analyzed for lactate concentration using a YSI (Yellow Springs, OH) 7100MBS. Lactate secretion was normalized to protein content at the end of the 6 hr treatment and is presented as relative to the 0 μM IA treatment.

## Measurement of mitochondrial and glycolytic stress responses

XF24 cell culture plates, sensor cartridges (100867–100) and XF base medium (103334–100) were purchased from Seahorse Bioscience (North Billerica, MA). Cells were seeded in XF24 cell culture plates at a density determined by optimization experiments and incubated at 37° C with 5% $CO_2$ overnight in growth medium; even distribution of cells was verified visually. For the mitochondrial stress test, growth medium was completely removed 24 hr after plating, and cells were washed twice with 1,000 μl of pre-warmed assay medium (XF base medium supplied with 25 mM glucose, 1 mM pyruvate and 2 mM glutamine; pH 7.4). 500 μl of assay medium was added to each well and cells were incubated in a 37°C incubator without $CO_2$ for 1 hr to allow cell equilibration with assay medium. Oxygen consumption rates were measured with the XF24 analyzer under this basal condition followed by sequential addition of 1 μM oligomycin, 4 μM FCCP, 1 μM rotenone, and 0.1 μM antimycin A. For the glycolytic stress test, growth medium was completely removed 24 hr after plating and replaced with glycolytic assay medium (XF base medium supplied with 5 mM glucose, 1 mM pyruvate, and 2 mM glutamine; pH 7.4) as described earlier. Extracellular acidification rates were measured under the basal condition, followed by sequential addition of 5 mM glucose, 1 μM oligomycin, and 50 mM 2-deoxyglucose (2-DG)(Nicholls et al., 2010).

## Replicates

Numbers of independent replicates are indicated in each figure legend as 'N'; we define 'independent replicate' here as a complete, separate performance of a time lapse imaging experiment with similar culture and treatment conditions, beginning from the plating of cells from bulk culture on an imaging plate and occurring on different days from other replicates. For all experiments shown, a minimum of 100 cells (not including daughters of cell divisions) were imaged and tracked for each condition. When possible, data from independent replicates were merged into a single data set using a set of calibration conditions for normalization of reporter signals; these data are indicated as 'combined' in the figure legends. When calibration controls were not available, comparisons among conditions were made within replicate experiments, and we verified that similar trends were observable in every replicate; these data are indicated as 'representative'. Determinations of statistical significance between distributions of pulse scores were made using the Kruskal-Wallace test, with the Bonferroni correction for multiple comparisons where applicable. All differences indicated with an asterisk were significant at the $p < 0.05$ level. Unless noted otherwise, where single-cell recordings are shown, the displayed cells were chosen by random number generation in MATLAB with a threshold for minimum tracking time to eliminate cells in which recording was terminated prematurely due to failure of the tracking algorithm, and the chosen tracks were manually verified to be representative of successfully tracked cells and consistent with the overall range of cell behaviors. Cell recordings determined by manual inspection to have poor tracking or quantification accuracy were discarded.

## Acknowledgements

Imaging facilities were provided by the Nikon Imaging Center at Harvard Medical School and the Cell Biology Imaging Facility at UC Davis. We thank P Tsou and L Cantley for providing the AMPKAR plasmid; G Gao, J Locasale, and T Muranen for providing reagents; D Clapham, T Schwarz, V Mootha, M Vander Heiden, S Gaudet, I Harris, and members of the Albeck laboratory, Brugge laboratory, and the Yellen laboratory for their comments. This work was supported by a Stuart H Q and Victoria Quan predoctoral fellowship (to YPH), a U.S. Department of Defense Breast Cancer Research Program postdoctoral fellowship (W81XWH-08-1-0609 to JGA), and the US National Institutes of Health (5-R01-CA105134-07 to JSB; R01 NS055031 to GY).

## Additional information

### Funding

| Funder | Grant reference number | Author |
|---|---|---|
| Congressionally Directed Medical Research Programs | W81XWH-08-1-0609 | John G Albeck |
| National Cancer Institute | 5-R01-CA105134-07 | Joan S Brugge |
| National Institute of Neurological Disorders and Stroke | R01-NS055031 | Gary Yellen |
| Stuart HQ and Victoria Quan | Predoctoral Fellowship | Yin P Hung |

The funders had no role in study design, data collection and interpretation, or the decision to submit the work for publication.

### Author contributions

Yin P Hung, Conceptualization, Data curation, Formal analysis, Validation, Investigation, Visualization, Writing—review and editing; Carolyn Teragawa, Data curation, Investigation, Project administration, Writing—review and editing; Nont Kosaisawe, Data curation, Software, Investigation, Visualization; Taryn E Gillies, Data curation, Software, Formal analysis, Visualization; Michael Pargett, Resources, Software, Validation, Methodology, Writing—review and editing; Marta Minguet, Data curation, Investigation, Project administration; Kevin Distor, Data curation, Software, Visualization, Project administration; Briana L Rocha-Gregg, Data curation, Formal analysis, Investigation, Visualization; Jonathan L Coloff, Data curation, Investigation, Methodology, Writing—review and editing; Mark A Keibler, Resources, Methodology, Writing—review and editing; Gregory Stephanopoulos, Resources, Supervision, Funding acquisition, Methodology, Writing—review and editing; Gary Yellen, Joan S Brugge, Conceptualization, Resources, Supervision, Funding acquisition, Investigation, Project administration, Writing—review and editing; John G Albeck, Conceptualization, Resources, Data curation, Software, Formal analysis, Supervision, Funding acquisition, Validation, Investigation, Visualization, Methodology, Writing—original draft, Project administration, Writing—review and editing

### Author ORCIDs

Yin P Hung, http://orcid.org/0000-0002-8568-1591
Gary Yellen, http://orcid.org/0000-0003-4228-7866
John G Albeck, http://orcid.org/0000-0003-2688-8653

### Decision letter and Author response

Decision letter https://doi.org/10.7554/eLife.27293.036
Author response https://doi.org/10.7554/eLife.27293.037

## Additional files

### Supplementary files

• Source Code 1. MATLAB files used to analyze peak size and frequency. Fluctuation scores were calculated on a cell-by-cell basis using the function ct_pulseanalysis. Unless otherwise noted, the parameter 'narm' was set to 3 and 'smooth' was set to 0. Subfunctions ct_getpeaks and ct_filter are required for ct_pulseanalysis, as is the MATLAB Signal Processing Toolbox.
DOI: https://doi.org/10.7554/eLife.27293.033

• Transparent reporting form
DOI: https://doi.org/10.7554/eLife.27293.034

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
