## [Decision Letter]

Thank you for submitting your article "Akt regulation of glycolysis mediates bioenergetic stability in epithelial cells" for consideration by *eLife*. Your article has been favorably evaluated by Jonathan Cooper (Senior Editor) and three reviewers, one of whom is a member of our Board of Reviewing Editors. The reviewers have opted to remain anonymous.

The reviewers have discussed the reviews with one another and the Reviewing Editor has drafted this decision to help you prepare a revised submission.

Summary:

Albeck and colleagues use a panel of biosensors to investigate the dynamics of metabolic homeostasis in single proliferating epithelial cells in response to a variety of pertubagens. Reporters for AMPK activation, Akt activation and NADH abundance were examined in single cells and populations, over time periods spanning several hours. All three probes reported predictable responses to major metabolic stressors, and some perturbations were accompanied by oscillatory responses in the metabolic/signaling state, many of which persisted for hours. Comparing the activity of multiple reporters simultaneously in the same cell suggested coordinated, inverse oscillations of Akt and AMPK. Overall, the authors find that when cells lack access to signals promoting metabolic flux and/or fuel to supply the relevant pathways, they enter into a prolonged pattern of pulsatile metabolic activity mediated by anti-synchronized cycles of AMPK and Akt. The attempt to resolve patterns of metabolic regulation at the single cell level is the main novelty of this work and addresses a key challenge in the field of cell metabolism. Although all three reviewers found some aspects of the work interesting, a number of concerns about the relevance of the study were raised.

Essential revisions:

1) The authors provide a great deal of information about cellular responses to stresses, but need to do more to tie these findings together in a way that will effectively communicate the core concepts to a broad audience. One approach would be to simplify the Results and Discussion around the key concept that glucose availability and insulin signaling are major drivers of the ability to achieve energetic stability in stressed cells.

2) The authors need to provide evidence that introduction of the probes does not affect glycolysis and its homeostatic regulation per se compared to parental cells.

3) Are these responses common to all epithelial cells or are they an idiosyncrasy of MCF10A cells? Generating a small subset of results (e.g. induction of oscillations in response to IA) in another cell line would inform on reproducibility and robustness.

4) A key concern is that it is unknown how relevant these fine-tuned oscillations are given the non-physiological conditions used. Clearly the oscillations are most dramatic under very harsh conditions (IA, oligomycin, etc.). A comparison with more therapeutically or physiologically relevant compounds such as metformin might be informative and would enhance the overall relevance of the paper to a broad audience. Furthermore, more should be done to quantify the metabolic responses to some of the inhibitors. The most obvious place to do this would be in the IA experiments. IA inhibits GAPDH, limiting both glycolysis and NADH production, yet the data suggest that glycolysis is required for oscillations in the NADH/NAD^+^ ratio. It would be helpful to know, for each of the IA concentrations used, how much glycolytic flux persists in a population of cells. Measuring glucose uptake is not adequate – ideally, the authors would use a radioisotope or stable isotope assay to measure flow through glycolysis (e.g. measuring conversion of 13C-glucose to 13C-lactate). The NADH/NAD^+^ ratio should also be measured across the population of cells for each IA concentration.

5) The kinetics of AMPK oscillation suggests that it could be driving NADH production. An obvious but unexplored mechanism is that AMPK induces fatty acid oxidation, which produced NADH. Does etomoxir or CPT1 silencing break the link between AMPK activation and NADH production?

[Editors' note: further revisions were requested prior to acceptance, as described below.]

Thank you for submitting your article "Akt regulation of glycolysis mediates bioenergetic stability in epithelial cells" for consideration by *eLife*. Your article has been reviewed by one peer reviewer, and the evaluation has been overseen by a Reviewing Editor and Jonathan Cooper as the Senior Editor.

Although all referees felt that the revision was markedly improved, they also felt that the issue of whether the sensors are influencing metabolism at the cellular level was not addressed thoroughly enough. The revision does compare metabolism between parental MCF10A cells and cell lines expressing the various sensors, but the reviewers found this experiment insufficient for two reasons.

First, it did not examine the influence of the sensors on metabolism at the cellular level, and specifically on metabolic oscillations in single cells; and second, although the cell lines have qualitatively similar responses to perturbations in the Seahorse experiment, the data do show a substantial metabolic impact for some of the probes, both at the level of basal metabolic rates and in the amplitude of some of the responses. Therefore it is still possible that the sensors themselves contribute to the metabolic activities they were designed to report.

A consultation including the Reviewer, Reviewing Editor and Senior Editor concluded that the paper could be considered again after one additional round of revision that more thoroughly addresses the question of the influence of sensor expression on metabolism. One way to address this concern would be to correlate expression levels/concentrations of both the AMPK and Akt sensors with oscillation frequency. Presumably you would be able to extract these data from previous live-cell imaging experiments, therefore allowing you to revise the paper quickly.

---

## [Author Response]

Essential revisions:1) The authors provide a great deal of information about cellular responses to stresses, but need to do more to tie these findings together in a way that will effectively communicate the core concepts to a broad audience. One approach would be to simplify the Results and Discussion around the key concept that glucose availability and insulin signaling are major drivers of the ability to achieve energetic stability in stressed cells.

We have edited the Results and Discussion section to better highlight the concept of insulin/glucose signaling as a mediator of energetic stability. We have simplified the results, summarizing the effects of other conditions more briefly (e.g., glutamine and pyruvate effects), and have rearranged the Discussion to put the main points earlier. Figure 5 has been moved to the main figures, and Figure 5—figure supplement 5D has been moved to the supplemental figures to best reflect the emphasis on the role of insulin signaling. We agree that these changes will better communicate the main point of the paper to a broader audience.

2) The authors need to provide evidence that introduction of the probes does not affect glycolysis and its homeostatic regulation per se compared to parental cells.

We now provide data using Seahorse measurements of extracellular acidification rate and oxygen consumption in response to metabolic perturbations (Figure 1—figure supplement 4), a standard measure of overall metabolic function that is often used to compare cell lines. These data indicate that the reporter cell lines do not deviate substantially from the parental cells in their response to metabolic stresses.

3) Are these responses common to all epithelial cells or are they an idiosyncrasy of MCF10A cells? Generating a small subset of results (e.g. induction of oscillations in response to IA) in another cell line would inform on reproducibility and robustness.

We now provide data from MCF12A and 184A1 (independently derived mammary epithelial cell lines) that show similar kinetics to MCF10A in response to the same perturbagens (Figure 3—figure supplement 2).

4) A key concern is that it is unknown how relevant these fine-tuned oscillations are given the non-physiological conditions used. Clearly the oscillations are most dramatic under very harsh conditions (IA, oligomycin, etc.). A comparison with more therapeutically or physiologically relevant compounds such as metformin might be informative and would enhance the overall relevance of the paper to a broad audience. Furthermore, more should be done to quantify the metabolic responses to some of the inhibitors. The most obvious place to do this would be in the IA experiments. IA inhibits GAPDH, limiting both glycolysis and NADH production, yet the data suggest that glycolysis is required for oscillations in the NADH/NAD^+^ ratio. It would be helpful to know, for each of the IA concentrations used, how much glycolytic flux persists in a population of cells. Measuring glucose uptake is not adequate – ideally, the authors would use a radioisotope or stable isotope assay to measure flow through glycolysis (e.g. measuring conversion of 13C-glucose to 13C-lactate). The NADH/NAD^+^ ratio should also be measured across the population of cells for each IA concentration.

We now provide data examining metformin treatment, and show that it induces AMPK activity fluctuations on a similar timescale to oligomycin treatment at concentrations where metformin is known to act as an inhibitor of the electron transport chain (Figure 3—figure supplement 2). We also show now that antimycin produces similar AMPK activity kinetics to both oligomycin and metformin, suggesting that this pattern of AMPK induction is common to inhibitors of mitochondrial ATP production. Because inhibition of the ETC or of OxPhos can occur in many physiologically and pathologically relevant situations, we think these new data help to increase the broader interest of the paper.

We also now include multiple experiments, including radioisotope tracing of glycolytic products, quantifying the glycolytic flux under high IA, low IA, and untreated conditions (Figure 4—figure supplement 2). These data all indicate that at 10 μM IA where oscillations are strongest, glycolytic flux is intermediate, supporting the conclusion that this treatment slows but does not eliminate flux through glycolysis.

We have also performed the experiment to measure NADH/NAD+ ratio across the cell population, and we show these results in Author response image 1. The changes observed with IA treatment occur in a similar direction to the measurements made by the reporter. However, there is a difficulty in comparing these experiments, as the bulk assay includes all of the subcellular pools of NADH and NAD^+^, while the reporter only indicates the cytosolic/nuclear pool. We believe that the fact that mitochondrial NADH accounts for a large fraction of the total NADH explains the relatively small change in NADH/NAD^+^ ratio in the population assay. Because of this caveat, the modest effect size, and the more conclusive glycolytic flux measurements made by 13C labeling, we have chosen not to include this panel in the current version of the paper. However, we would be happy to include this panel if the reviewers feel that it would add significantly to the paper.

**Author response image 1. respfig1:** Population measurement of NADH/NAD^+^ ratio in response to IA treatment. Measurements of NADH and NAD^+^ were made on MCF10A cells following treatment with the indicated compounds for 2 hr, using the NAD/NADH Glo Assay (Promega) and the protocol of Sullivan et al., 2015 (PMID: 26232225).

5) The kinetics of AMPK oscillation suggests that it could be driving NADH production. An obvious but unexplored mechanism is that AMPK induces fatty acid oxidation, which produced NADH. Does etomoxir or CPT1 silencing break the link between AMPK activation and NADH production?

This is a very interesting point and we have now examined the effect of etomoxir on IA-induced oscillations. Similar to PI3K inhibitors, we find that high concentrations of etomoxir dampen but do not eliminate oscillations (Figure 4—figure supplement 3). This result is consistent with our model that a multi-pronged feedback structure is responsible for the oscillations.

[Editors' note: further revisions were requested prior to acceptance, as described below.]

[…] A consultation including the Reviewer, Reviewing Editor and Senior Editor concluded that the paper could be considered again after one additional round of revision that more thoroughly addresses the question of the influence of sensor expression on metabolism. One way to address this concern would be to correlate expression levels/concentrations of both the AMPK and Akt sensors with oscillation frequency. Presumably you would be able to extract these data from previous live-cell imaging experiments, therefore allowing you to revise the paper quickly.

We thank the reviewers and editors for their positive assessment, and for the suggestions to further improve the manuscript with evidence that our reporters do not impact the observed behavior. We agree that this is an important caveat to consider, and we have now addressed it in two ways:

1) We have performed the suggested analysis, which is now presented in Figure 3—figure supplement 1. Using a dataset of >3,000 cells we have divided the cells into bins by expression of either AMPKAR2 or Akt-KTR, and have calculated the distribution of pulse parameters within each of these subpopulations. We also show representative single-cell traces from the bottom-10 and top-10 percentiles of expression for each reporter. The result of this analysis is that the kinetics of oscillatory behavior are qualitatively similar across all reporter expression levels. Quantitatively, we observe a moderate trend toward shorter and more frequent pulses in cells with higher reporter expression. While this trend could suggest that reporter expression influences dynamics to a limited extent, the effect size is substantially weaker than the other effects we report in our study and is not strong enough to suggest that reporters severely distort the dynamics of the observed metabolic responses. It is also worth noting an alternative explanation for this correlation, which is that the cell-specific factors leading to higher reporter expression in certain cells may also modify their metabolic behavior.

2) We have taken a new experimental approach, presented in Figure 3—figure supplement 2, in which we directly compare reporter-expressing and non-reporter-expressing (parental) MCF10A cells under the same conditions, performing an immunofluorescence time course to detect phosphorylation of the physiological AMPK target ACC. Because ACC and the AMPKAR2 reporter are both direct targets of AMPK, we expect these immunofluorescence measurements to be very similar to the reporter data shown in Figure 3, albeit at fixed time points. As shown in the images, for both parental and reporter cells we do in fact observe an increase in pACC-positive cells after 1 hour of oligomycin treatment, followed by a decrease at 2 hours, and then an increase at 3 hours, mirroring the average behavior measured by AMPKAR2 (Figure 3). Importantly, in both cell types examined, after >3 hours of treatment with either IA or oliogmycin we observe cells with both very strong and very weak pACC staining, which is exactly the pattern that would be expected in asynchronously oscillating cells measured at a fixed time point. The similarity of pACC patterns between parental and reporter cells, and to live-cell AMPKAR measurements, further supports the conclusion that the reporters do not substantially alter the drug-induced dynamics of AMPK activity.

Together, we think that these new analyses adequately establish that expression of reporters is effectively neutral for the effects being studied. We have rewritten the relevant parts of the Results section to highlight our efforts to establish that our observations are not specific to the experimental system and methodology used.

We also note here that we have made a correction to our pulse analysis software; in the process of performing the re-analysis above, we discovered a bug which caused some peaks to be missed and reduced the accuracy of the scoring. We have updated all plots showing pulse scores accordingly; the affected plots are Figure 3, Figure 3—figure supplement 3, Figure 4, Figure 4—figure supplement 1, Figure 4—figure supplement 3, Figure 4—figure supplement 3, Figure 5, Figure 5, Figure 6, Figure 6, Figure 7, and Figure 7. The changes are very minor and none of the conclusions or statistical comparisons are affected, but we feel that it is important to use the most accurate analysis possible and to be forthright with any changes to the manuscript.